# The Predictive-Corrective Paradigm: Decoupling Prediction and Refinement for Efficient Anatomy-Informed Brain MRI Segmentation

## Abstract

In medical image segmentation, although end-to-end deep learning has achieved substantial progress, obtaining accurate results typically requires many training iterations and large-scale annotated datasets, which limits efficiency and practicality in data-scarce clinical scenarios. To address this issue, we propose a Predictive–Corrective (PC) paradigm that decouples segmentation into a fast, anatomy-informed prediction stage followed by a focused refinement stage. Based on this paradigm, we develop PCMambaNet, which comprises two cooperative modules: a Predictive Prior Module (PPM) that generates a coarse anatomical approximation at low computational cost and injects symmetry priors via inter-hemispheric similarity and thresholding to highlight diagnostically relevant asymmetric regions, and a Corrective Residual Network (CRN) that models the residual error, concentrating capacity on refining challenging regions and delineating pathological boundaries. Experiments on multiple high-resolution brain MRI benchmarks show that PCMambaNet attains competitive accuracy with relatively few training epochs and exhibits clear advantages in data-limited settings. Extended experiments further indicate that the proposed PC paradigm remains applicable to organs without strong left–right symmetry. Overall, this work demonstrates that explicitly incorporating anatomy-informed priors and decoupling prediction from refinement is an effective way to improve both training efficiency and data efficiency in medical image segmentation.

## 1 Introduction

In medical image segmentation, the end-to-end learning paradigm (Rayed et al., 2024; Zhang et al., 2024b), especially when implemented with Convolutional Neural Networks (CNNs) (Ronneberger et al., 2015) and Transformers (Rahman et al., 2024; Cao et al., 2022), has achieved remarkable success in computer vision and is widely used in medical image analysis, improving the accuracy of diagnosis and treatment planning (Chen et al., 2024b). Yet a fundamental paradox remains: prevailing end-to-end models essentially follow a "brute-force" strategy, where a single monolithic network (Long et al., 2015) is asked to directly learn a highly non-linear mapping from raw input to final output. This all-in-one design is flexible but often incurs high training and inference costs, a strong reliance on large labeled datasets, and heavy computational demands (LeCun et al., 2015; He et al., 2016). In the medical domain—where data are scarce, annotation is costly, and efficiency and robustness are critical—this demand for data and computation has become a major bottleneck to further progress and clinical deployment (Dong et al., 2024). Figure 1 illustrates this behavior on two benchmarks by plotting validation-loss trajectories and inter-epoch parameter $L_2$ distances for several representative models, showing that strong performance for standard end-to-end baselines typically comes at the price of long training schedules and many small parameter updates.

To mitigate these limitations, numerous strategies have been explored (Zheng et al., 2024). Transfer learning pre-trains models on large public datasets such as ImageNet (Matsoukas et al., 2022), but cannot remove the dependence on target-domain annotations (Cheplygina et al., 2019). Advanced data augmentation enlarges the effective dataset, yet may introduce artifacts and distribution shifts (Fabian et al., 2021). More powerful architectures, including Vision Transformers (ViTs) (Dosovitskiy et al., 2020) and State Space Models (SSMs) (Gu & Dao, 2023; Zhu et al., 2024), further

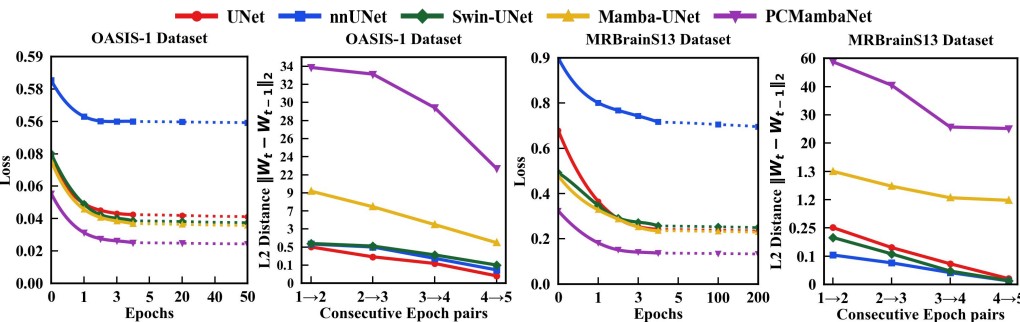

Figure 1: The efficacy of the Predictive–Corrective (PC) paradigm. We compare PCMambaNet with representative end-to-end baselines in terms of validation-loss trajectories (first and third plots) and inter-epoch parameter $L_2$ distances between consecutive epoch pairs (second and fourth plots) on the OASIS-1 and MRBrainS13 datasets. Compared with standard end-to-end models, PCMambaNet shows a faster reduction in validation loss and larger parameter updates in the early training stage. These observations indicate that reformulating the task as a predictive prior plus a corrective residual, guided by anatomy-informed priors, can lead to more efficient use of training data and optimization budget during training.

boost performance, but do not change the brute-force nature of end-to-end learning and can even exacerbate data and computation requirements (Liu et al., 2024a). These approaches mainly improve how we fit the model to the data, without simplifying the underlying learning problem itself. This naturally raises a key question: instead of asking a single network to solve everything at once, can we reformulate medical image segmentation as a prediction–correction process that leverages anatomy-informed priors to make learning more efficient and more data-efficient?

To answer this, we propose a **Predictive–Corrective (PC) paradigm** that decomposes segmentation into two tractable stages. A lightweight **Predictive Prior Module (PPM)** first generates a coarse, anatomy-guided initial guess that narrows the search space. A subsequent **Corrective Residual Network (CRN)** then focuses on modeling the residual error to refine boundaries and difficult regions, effectively simplifying the overall learning objective.

We instantiate this paradigm on high-resolution brain MRI segmentation by exploiting the approximate left–right symmetry of the human brain. The PPM builds an inter-hemispheric similarity map and applies thresholding to generate a **focus map** highlighting asymmetric and diagnostically relevant areas. When symmetry is weak, the PPM adaptively produces a smoother, more conservative mask, avoiding overly strong prior bias. The CRN, implemented with dynamically density-weighted convolutions, refines details within these high-probability regions, naturally combining anatomical priors with deep representations.

Built on this design, we develop **PCMambaNet**. Experiments on multiple brain MRI benchmarks show that PCMambaNet achieves competitive or state-of-the-art performance while requiring fewer training epochs and exhibiting clear advantages under limited-data conditions. Additional experiments on cardiac segmentation—an organ without strong bilateral symmetry—confirm the generality of the proposed paradigm. The main contributions of this work can be summarized as follows:

- We introduce the Predictive-Corrective (PC) paradigm, a novel framework designed to substantially enhance data efficiency by simplifying the learning objective through predictive priors and residual correction.

- We demonstrate how to successfully instantiate this paradigm in a challenging medical segmentation task by leveraging anatomical prior knowledge.

- We show that PCMambaNet attains state-of-the-art performance with substantially fewer training epochs, mitigating issues related to data inefficiency and overfitting.

## 2 METHOD

### 2.1 ARCHITECTURE OVERVIEW

Our method is built upon the previously introduced **Predictive–Corrective (PC) paradigm**, which reformulates medical image segmentation as two explicitly decoupled stages: a fast, anatomy-informed prediction stage and a subsequent residual refinement stage. This design alleviates the limitations of conventional monolithic end-to-end modeling by simplifying the learning objective and reducing reliance on large-scale labeled datasets; a conceptual discussion of this decomposition is provided in Appendix D. To instantiate this paradigm, we propose **PCMambaNet**, a U-shaped segmentation network whose core component is the newly designed **PCMamba module**. As illustrated in Figure 2, each PCMamba module realizes the PC principle through two parallel branches: a **Predictive Prior Module (PPM)** that generates a coarse, anatomy-guided prediction, and a **Corrective Residual Network (CRN)** that focuses on refining residual errors and delineating fine structural boundaries. The outputs of these two branches are then fused within the module to form a unified representation, which is further propagated through the network for accurate segmentation.

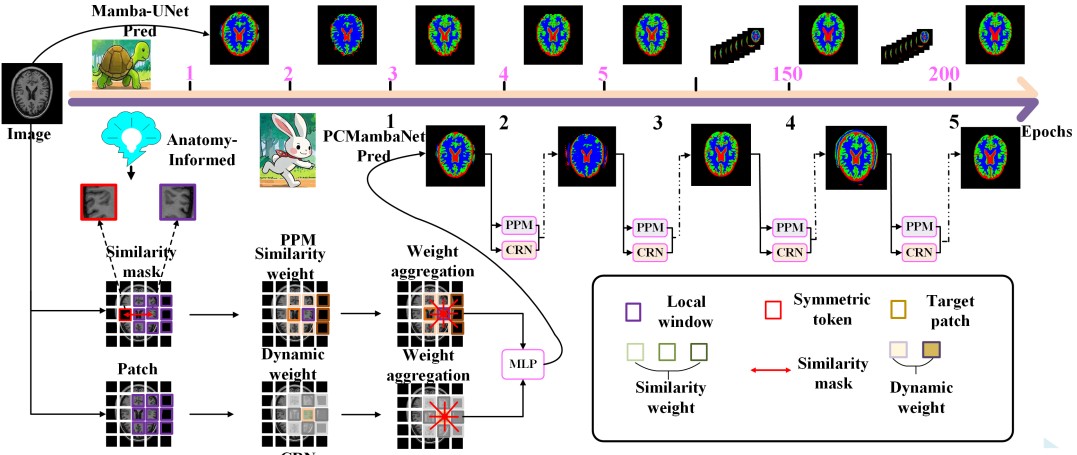

Figure 2: Architecture of **PCMambaNet**, which instantiates the proposed **Predictive–Corrective (PC) paradigm**. Each module contains two branches: a **Predictive Prior Module (PPM)** producing a coarse anatomy-guided prediction, and a **Corrective Residual Network (CRN)** refining residual errors and boundaries. This decoupled design enables efficient training and strong performance under full and limited data.

### 2.2 PRELIMINARIES

State space models (SSMs) are widely employed for analyzing sequential data and modeling continuous linear time-invariant (LTI) systems (Gu & Dao, 2023). Given an input sequence $u(t) \in \mathbb{R}$, the system maps it to an output sequence $y(t) \in \mathbb{R}$ through the hidden state $x(t) \in \mathbb{C}^N$. Here, $t > 0$ denotes the time index and $N$ is the state dimension. The dynamics of the system can be described by the following state transition and observation equations:

$$\dot{x}(t) = \boldsymbol{A}x(t) + \boldsymbol{B}u(t), \qquad y(t) = \boldsymbol{C}\boldsymbol{x}(t) + \boldsymbol{D}u(t), \tag{1}$$

where $\boldsymbol{A} \in \mathbb{C}^{N \times N}$ is the state transition matrix, $\boldsymbol{B} \in \mathbb{C}^{N \times 1}$ and $\boldsymbol{C} \in \mathbb{C}^{1 \times N}$ are the input and output projection matrices, and $\boldsymbol{D} \in \mathbb{C}$ represents the skip connection. These equations specify how the hidden state evolves over time and how it relates to the observable output.

To integrate continuous-time SSMs into deep learning frameworks, it is necessary to discretize them. A common approach is the Zero-Order Hold (ZOH) discretization. Given the sampling interval $\boldsymbol{\Delta}$, the discrete system parameters can be expressed as

$$\bar{\boldsymbol{A}} = e^{\boldsymbol{\Delta}\boldsymbol{A}}, \qquad \bar{\boldsymbol{B}} = \boldsymbol{A}^{-1}\left(e^{\boldsymbol{\Delta}\boldsymbol{A}} - I\right)\boldsymbol{B}. \tag{2}$$

In practice, however, real-world processes are often time-varying and cannot be sufficiently modeled by a fixed LTI system. To address this limitation, improved SSM formulations allow the parameters to adapt to the input, thereby enhancing the modeling capacity. Specifically, the parameters $\boldsymbol{\Delta}$, $\boldsymbol{B}$, and $\boldsymbol{C}$ can be defined as functions of the input sequence $\{u_t\}$:

$$\boldsymbol{\Delta}_t = s_\Delta(u_t), \qquad \boldsymbol{B}_t = s_B(u_t), \qquad \boldsymbol{C}_t = s_C(u_t). \tag{3}$$

Based on these input-dependent parameters, the discrete dynamics can be written as

$$x_t = \bar{\boldsymbol{A}}_t x_{t-1} + \bar{\boldsymbol{B}}_t u_t, \qquad y_t = \boldsymbol{C}_t x_t + \boldsymbol{D} u_t, \tag{4}$$

where $\bar{\boldsymbol{A}}_t$ and $\bar{\boldsymbol{B}}_t$ are computed from the adaptive parameters.

## 2.3 PREDICTIVE-CORRECTIVE MAMBA BLOCK (PCMAMBA)

The **Predictive-Corrective Mamba (PCMamba)** block is the core of our method, designed to instantiate our PC paradigm within the state space model. Its central mechanism is to modulate the original Mamba state evolution by explicitly incorporating predictive priors and corrective local details. To achieve this, we introduce a novel state modulation equation into the original Mamba formulation.The overall architecture of the PCMamba block is illustrated in Appendix A.2, Figure 7 (b). Its workflow can be described by **contextual modulation factor $\boldsymbol{z}_t$** to the state equation:

$$x_t = \overline{\boldsymbol{A}}_t x_{t-1} + \overline{\boldsymbol{B}}_t u_t, \tag{5}$$
$$h_t = \boldsymbol{z}_t \odot x_t, \tag{6}$$
$$y_t = \boldsymbol{C}_t h_t + \boldsymbol{D} u_t. \tag{7}$$

Here, Eq. 5 is the standard Mamba state transition. Our key innovation lies in Eq. 6, where the original state $x_t$ is element-wise modulated by $\boldsymbol{z}_t$ to produce a new, context-aware state $h_t$. The observation $y_t$ is then computed from this modulated state. The modulation factor $\boldsymbol{z}_t$ is dynamically generated by two parallel branches that work in concert to implement our PC paradigm.

In practice, we implement the computation of the modulation factor $\boldsymbol{z}_t$ through two specialized branches: a Predictive Branch based on symmetric mask aggregation and a Corrective Branch based on dynamically density-weighted local modeling. The outputs of these two branches are fused to form the final modulation factor.

### 2.3.1 PREDICTIVE BRANCH: SYMMETRIC MASK AGGREGATION (PPM)

**This branch implements our Predictive Prior Module (PPM)**, whose core objective is to leverage the anatomical symmetry prior to automatically identify and focus on structurally anomalous regions, which are indicative of potential pathologies. The detailed architecture of the module is illustrated in Figure 2. This process is achieved in the following three steps:

**Step 1: Constructing the Comparative Feature Set.** For each spatial location $i$ in the feature map, we construct a feature set for comparison, $\mathcal{T}_i$. Critically, this set includes not only features from its local neighborhood $\mathcal{N}(i)$ but also the feature from its symmetric counterpart position $i'$, such that $\mathcal{T}_i = \{\mathbf{x}_j \mid j \in \mathcal{N}(i) \cup \{i'\}\}$.

**Step 2: Generating the Similarity Mask.** We quantify the structural difference between the center token $\mathbf{x}_i$ and each feature $\mathbf{x}_j$ in $\mathcal{T}_i$ by computing their cosine similarity, $s_{i,j}$. We select cosine similarity as it is invariant to the magnitude of the features, allowing it to measure differences in structure (i.e., direction) more purely. Subsequently, based on a predefined similarity threshold $\theta$, a binary mask is generated:

$$m_{i,j} = \mathbb{I}[s_{i,j} < \theta]. \tag{8}$$

This mask is designed to "filter out a-normalcy": when two regions are structurally similar ($s_{i,j} \geq \theta$), likely corresponding to healthy, symmetric tissue, the mask value is 0. Conversely, when a significant difference exists ($s_{i,j} < \theta$), indicating a potential disruption of symmetry by a lesion, the mask value is 1.

**Step 3: Aggregating Anomaly Features.** Finally, using the mask generated in the previous step, we aggregate only those features identified as "structurally anomalous" via a normalized weighted summation. This produces the final output of the predictive branch, $z_i^{\text{mask}}$. This process explicitly encodes the "where to look" prior, providing precise guidance for the subsequent corrective module.

### 2.3.2 CORRECTIVE BRANCH: DYNAMICALLY DENSITY-WEIGHTED LOCAL MODELING (CRN)

**This branch constitutes the core of our Corrective Residual Network (CRN)**, with its primary responsibility being fine-grained local detail modeling, which is essential for the precise delineation of lesion boundaries. The detailed architecture of the module is illustrated in Figure 2. This process is realized through the following steps:

**Step 1: Receptive Field Expansion and Local Feature Extraction.** For each spatial position $i$, we employ dilated convolution to extract a local feature patch $P_i$. We opt for dilated convolution as it effectively expands the receptive field without increasing computational cost or the number of parameters. This allows the model to capture a broader local context, which is critical for understanding complex tissue structures.

**Step 2: Dynamic Weight Generation.** The extracted local feature patch $P_i$ is flattened and fed into a lightweight Multilayer Perceptron (MLP) to dynamically generate an adaptive weight vector $\boldsymbol{\beta}_i$ for each pixel within that local region:

$$\boldsymbol{\beta}_i = \text{Softmax}\left(\text{MLP}\left(\text{Flatten}(P_i)\right)\right). \tag{9}$$

The Softmax function ensures the normalization of these weights. This weight vector, $\boldsymbol{\beta}_i$, can be interpreted as an attention map learned by the model based on the local content, indicating which pixels are more informative for an accurate segmentation.

**Step 3: Weighted Feature Aggregation.** Finally, we use the dynamic weights $\boldsymbol{\beta}_i$ generated in the previous step to perform a weighted summation of the pixels within the local feature patch, yielding a finely refined local representation, $p_i^{\text{d}}$. This representation is then passed through a linear mapping to produce the final output of the corrective branch, $z_i^{\text{density}}$. This process provides the model with crucial information on "how to refine details," serving as the perfect complement to the "where to look" guidance from the predictive branch.

## 3 EXPERIMENTS

In this section, we present a series of comprehensive experiments to validate the effectiveness of our proposed Predictive–Corrective (PC) paradigm. We systematically evaluate our approach by (1) comparing its segmentation accuracy and training dynamics with state-of-the-art (SOTA) methods, (2) analyzing the necessity of each component within the PC paradigm, namely the PPM and the CRN, (3) assessing its performance in data-scarce scenarios, and (4) demonstrating its qualitative advantages through visual analysis. The datasets used in the extended experiments are provided in Appendix B.1, and the corresponding detailed results are reported in Appendix B, Tables 10 and 11.

### 3.1 DATASETS

**OASIS-1:** The dataset used in this study is derived from the Open Access Series of Imaging Studies (OASIS) (Marcus et al., 2007) and is referred to as OASIS-1. It consists of data collected from 421 individuals, aged 18 to 96 years, each of whom underwent a T1-weighted magnetic resonance imaging (MRI) scan. The MRI acquisition parameters are as follows: TR (9.7 ms), TE (4.0 ms), flip angle (10°), TI (20 ms), TD (200 ms), with a slice thickness of 1.25 mm and a resolution of 176×208 pixels, without any gaps between slices. **MRBrainS13:** This dataset comprises multi-sequence brain MRI scans of 20 subjects acquired on a 3.0 T Philips Achieva scanner at the University Medical Center Utrecht, the Netherlands (Mendrik et al., 2015). The cohort includes older adults (age > 50 years) with cardiovascular risk factors, such as patients with type 2 diabetes mellitus and age- and sex-matched controls, and thus exhibits varying degrees of brain atrophy and white matter hyperintensities, while individuals with large territorial infarcts, major stroke, or other overt focal brain pathology were excluded. For each subject, T1-weighted, T1-weighted inversion recovery (T1-IR), and T2-FLAIR images were acquired and subsequently rigidly registered using Elastix (Klein et al., 2009) and bias-field corrected using SPM8 (Ashburner & Friston, 2005), resulting in a unified voxel spacing of $0.96 \times 0.96 \times 3.00$ mm.

## 3.2 Implementation Details

Our model is implemented in PyTorch, and all experiments are conducted on a single NVIDIA A40 GPU with 48 GB of memory under Ubuntu 22.04 and Python 3.10. We use the AdamW optimizer with an initial learning rate of $0.0001$, decayed using a cosine annealing schedule. The hyperparameter $\theta$ is empirically selected and fixed at 0.95 for all experiments; detailed hyperparameter sensitivity analysis is provided in Appendix C. Training is performed on 2D axial slices from two datasets, which are split into training, validation, and test sets in an $8 : 1 : 1$ ratio. Specifically, the training set of OASIS-1 contains 52,094 slices, and that of MRBrainS13 contains 768 slices. All input images are resampled to a resolution of $224 \times 224$, the batch size is set to 12, and no data augmentation is applied during training. Unless otherwise specified, models are trained for 50 epochs on large-scale datasets and 200 epochs on small-scale datasets, and the checkpoint achieving the highest Dice score on the validation set is used for final evaluation. The code will be released upon acceptance of the paper.

## 3.3 Comparison with State-of-the-Art Methods

To comprehensively evaluate the effectiveness of the proposed PCMambaNet, we compare it against several representative segmentation baselines, including classic CNN-based methods (U-Net (Ronneberger et al., 2015) and nnUNet (Isensee et al., 2021)), the Transformer-based model Swin-UNet (Cao et al., 2022), and the end-to-end Mamba-based model Mamba-UNet (Wang et al., 2024b). This comparative setup enables us to verify that the observed performance improvements stem from the predictive–corrective paradigm itself rather than being solely attributable to the Mamba architecture.

**Quantitative Results.** The quantitative results in Table 1 show that **PCMambaNet** achieves state-of-the-art segmentation performance while maintaining strong efficiency. On the large-scale OASIS-1 dataset, PCMambaNet trained for 50 epochs attains the best results across all tissues and metrics, and its HD95 and ASD scores are consistently lower than those of all baselines, indicating more accurate and stable boundary delineation. Moreover, even with only 5 epochs of training, PCMambaNet already reaches performance comparable to or better than several fully trained baselines, reflecting the benefits of the proposed Predictive–Corrective design for efficient optimization. On the small-scale MRBrainS13 dataset, the advantages become more pronounced. PCMambaNet achieves the best Dice, IOU, HD95, and ASD across all tissues and clearly outperforms the strongest CNN baseline, **UNet**. On average, PCMambaNet improves Dice and IOU by roughly **1–2** percentage points over UNet, while also yielding consistently lower HD95 and ASD. These results indicate that the proposed PC paradigm is particularly effective in limited-data regimes. These results, consistent across datasets of varying scales, validate the strong inductive bias and fine-grained modeling capabilities endowed by our Predictive-Corrective paradigm. A more comprehensive evaluation including additional metrics (Acc, Pre, Sen, Spe) is provided in Appendix A.4, Table 5.

Table 1: **PCMambaNet achieves state-of-the-art efficiency without compromising accuracy.** This table presents a quantitative comparison on the OASIS-1 and MRBrainS13 test sets. Our method (highlighted) matches state-of-the-art (SOTA) performance on accuracy metrics while consistently outperforming all competitors in terms of efficiency. Dice and IOU are reported as percentages, while HD95 and ASD are scaled by a factor of 10 for readability. Best results are in **bold**.

| Model | Dice (%) ↑ | | | HD95 (mm×10) ↓ | | | ASD (mm×10) ↓ | | | IOU (%) ↑ | | |
|---|---|---|---|---|---|---|---|---|---|---|---|---|
| | CSF | GM | WM | CSF | GM | WM | CSF | GM | WM | CSF | GM | WM |
| **OASIS-1 dataset** | | | | | | | | | | | | |
| UNet | 91.18 ± 0.61 | 92.43 ± 0.60 | 93.16 ± 0.23 | 12.54 ± 0.08 | 12.67 ± 0.08 | 19.66 ± 0.39 | 2.69 ± 0.09 | 2.95 ± 0.16 | 5.20 ± 0.15 | 85.43 ± 1.06 | 87.91 ± 0.98 | 89.20 ± 0.44 |
| nnUNet | 90.38 ± 0.01 | 92.09 ± 0.04 | 92.92 ± 0.02 | 14.17 ± 0.09 | 11.63 ± 0.03 | 17.77 ± 0.15 | 4.25 ± 0.01 | 3.07 ± 0.02 | 5.12 ± 0.07 | 83.99 ± 0.01 | 87.38 ± 0.04 | 88.65 ± 0.02 |
| Swin-UNet | 90.98 ± 0.51 | 92.43 ± 0.32 | 92.57 ± 0.05 | 13.53 ± 0.49 | 12.02 ± 0.31 | 19.44 ± 1.37 | 3.88 ± 0.32 | 3.08 ± 0.01 | 4.82 ± 0.28 | 85.05 ± 0.91 | 87.92 ± 0.48 | 88.76 ± 0.11 |
| Mamba-Unet | 91.95 ± 0.28 | 92.89 ± 0.06 | 93.31 ± 0.32 | 11.60 ± 0.20 | 11.78 ± 0.24 | 18.75 ± 0.84 | 2.74 ± 0.25 | 2.76 ± 0.22 | 4.26 ± 0.26 | 86.73 ± 0.56 | 88.77 ± 0.36 | 89.25 ± 0.64 |
| **PCMambaNet(1 epoch)** | 91.21 ± 0.25 | 91.78 ± 0.03 | 92.03 ± 0.24 | 11.72 ± 0.02 | 13.53 ± 0.06 | 29.72 ± 0.04 | 3.41 ± 0.05 | 3.85 ± 0.35 | 5.36 ± 0.04 | 83.75 ± 0.04 | 86.72 ± 0.21 | 86.31 ± 0.24 |
| **PCMambaNet(5 epochs)** | 92.58 ± 0.25 | 92.39 ± 0.52 | 92.91 ± 0.34 | 10.80 ± 0.03 | 11.49 ± 0.26 | 18.32 ± 0.26 | 2.10 ± 0.03 | 2.53 ± 0.06 | 3.55 ± 0.08 | 87.38 ± 0.65 | 88.17 ± 0.46 | 89.45 ± 0.37 |
| **PCMambaNet(50 epochs)** | **94.10 ± 0.38** | **94.33 ± 0.35** | **94.29 ± 0.34** | **10.58 ± 0.03** | **10.81 ± 0.17** | **15.00 ± 1.25** | **1.76 ± 0.02** | **1.88 ± 0.18** | **2.70 ± 0.38** | **89.91 ± 0.08** | **91.22 ± 0.50** | **92.23 ± 0.66** |
| **MRBrainS13 dataset** | | | | | | | | | | | | |
| UNet | 67.68 ± 0.71 | 70.58 ± 0.38 | 73.80 ± 0.64 | 22.77 ± 0.55 | 20.02 ± 0.99 | 43.99 ± 2.63 | 4.67 ± 0.07 | 4.71 ± 0.06 | 16.10 ± 2.04 | 58.79 ± 0.82 | 63.18 ± 0.58 | 66.36 ± 0.75 |
| nnUNet | 62.36 ± 1.69 | 65.79 ± 1.32 | 66.17 ± 1.18 | 42.69 ± 0.22 | 50.89 ± 0.21 | 74.02 ± 1.85 | 9.38 ± 1.27 | 15.45 ± 2.92 | 54.30 ± 2.58 | 50.38 ± 2.62 | 56.27 ± 1.96 | 58.14 ± 1.78 |
| Swin-UNet | 64.43 ± 1.87 | 68.66 ± 0.62 | 69.70 ± 1.11 | 28.31 ± 4.60 | 35.68 ± 1.19 | 59.97 ± 3.00 | 6.89 ± 1.15 | 10.63 ± 1.15 | 23.37 ± 5.71 | 54.23 ± 2.49 | 60.48 ± 0.91 | 62.32 ± 1.03 |
| Mamba-Unet | 64.86 ± 0.40 | 68.99 ± 0.46 | 70.57 ± 0.52 | 28.38 ± 0.39 | 33.17 ± 3.93 | 52.80 ± 4.08 | 6.18 ± 0.13 | 9.35 ± 1.16 | 17.43 ± 3.45 | 54.74 ± 0.46 | 60.81 ± 0.52 | 63.01 ± 0.64 |
| **PCMambaNet(1 epoch)** | 56.47 ± 0.45 | 60.12 ± 0.03 | 51.15 ± 0.33 | 48.61 ± 0.03 | 51.56 ± 0.10 | 99.66 ± 0.45 | 9.08 ± 0.06 | 17.05 ± 0.59 | 24.75 ± 1.25 | 44.91 ± 0.42 | 48.22 ± 0.28 | 41.04 ± 2.86 |
| **PCMambaNet(5 epochs)** | 62.87 ± 0.55 | 65.97 ± 2.64 | 64.18 ± 0.34 | 29.98 ± 0.67 | 39.03 ± 0.53 | 45.26 ± 0.43 | 6.73 ± 0.88 | 11.96 ± 0.48 | 14.34 ± 0.59 | 51.26 ± 5.97 | 55.76 ± 6.77 | 56.91 ± 2.23 |
| **PCMambaNet(200 epochs)** | **69.37 ± 0.41** | **71.79 ± 0.80** | **74.26 ± 0.93** | **19.31 ± 1.98** | **17.46 ± 2.00** | **38.90 ± 7.33** | **4.47 ± 0.23** | **4.60 ± 0.51** | **11.41 ± 2.18** | **60.67 ± 0.97** | **64.63 ± 1.14** | **67.48 ± 1.08** |

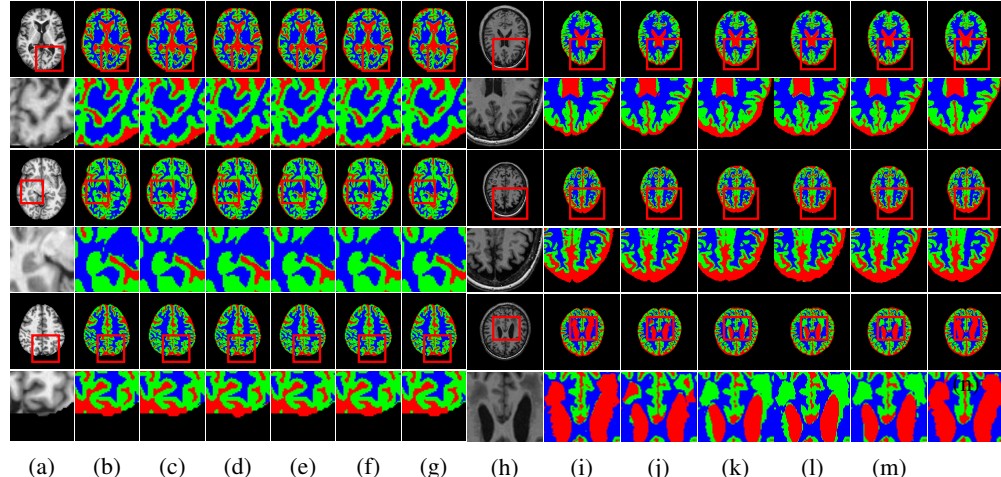

(a) (b) (c) (d) (e) (f) (g) (h) (i) (j) (k) (l) (m)

Figure 3: Qualitative comparison of segmentation results on the OASIS-1 and MRBrainS13 datasets. For the OASIS-1 dataset, the columns are: (a) original image, (b) ground-truth (GT) annotation, (c) UNet, (d) nnUNet, (e) Swin-UNet, (f) Mamba-UNet, and (g) PCMambaNet (ours). For the MRBrainS13 dataset, the columns are: (h) original image, (i) ground-truth (GT) annotation, (j) UNet, (k) nnUNet, (l) Swin-UNet, (m) Mamba-UNet, and (n) PCMambaNet (ours). The proposed **PCMambaNet** (last column in each group) produces segmentations that are more consistent with the ground truth in terms of boundary delineation, structural coherence, reduced false positives, and finer depiction of complex anatomical regions. The zoomed-in regions within red boxes highlight representative areas where the advantages of our method are particularly evident.

**Qualitative Results.** The qualitative results shown in Figure 3 visually corroborate the aforementioned quantitative findings and further highlight the architectural advantages of **PCMambaNet**. On the large-scale dataset, baseline models often struggle to accurately capture fine boundaries, tending to produce noisy or incomplete segmentations. In contrast, our model leverages the **PPM** to focus on critical regions, thereby enabling the **CRN** to delineate complex contours that are highly consistent with the ground truth. The superiority of our method becomes even more pronounced under data-limited conditions (small-scale datasets): competing models typically generate overly smooth results that fail to capture fine-grained, infiltrative structures, whereas our "predict-then-correct" strategy effectively recovers these complex boundary details. Consequently, **PCMambaNet** yields visually sharper and more anatomically plausible segmentation results, particularly in challenging low-data scenarios.

**Training dynamics.** Figure 4 compares the validation Dice trajectories of PCMambaNet and representative end-to-end baselines. To make the optimization behaviour comparable, all models are trained with the same relatively small learning rate. Under this setting, the validation Dice of **PCMambaNet** increases more rapidly in the early stage and reaches a stable high-performance regime within roughly the first **5–10 epochs**. In contrast, baseline end-to-end models such as U-Net, Swin-UNet, and Mamba-UNet improve more gradually and require substantially more epochs to approach their best performance. These observations suggest that reformulating the task under the PC paradigm enables more effective use of early optimization steps and leads to more favourable training dynamics.

### 3.4 ABLATION STUDY

As shown in Table 2, the ablation results on OASIS-1 and MRBrainS13 indicate that the PPM and CRN are both indispensable and complementary. Removing the PPM or replacing it with a random mask significantly degrades Dice/IOU and boundary metrics, especially on the small-scale MRBrainS13 dataset, confirming that the PPM provides an effective inductive bias under limited data. Likewise, removing or weakening the CRN consistently harms performance, demonstrating that a high-capacity, carefully designed refinement module is critical to the superiority of the PC

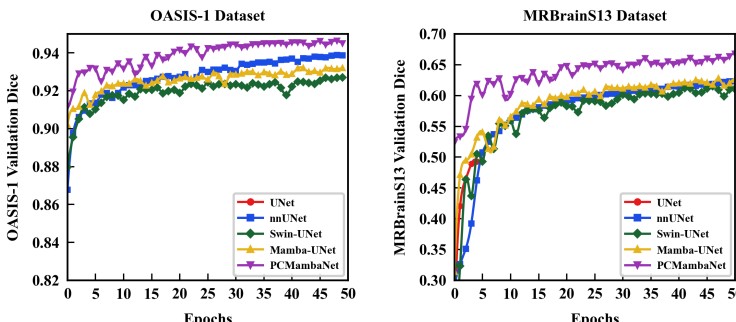

Figure 4: **PCMambaNet enables much faster training in practice.** The validation Dice score of our model (blue curve) reaches near-optimal performance within the first **5 epochs**. In contrast, conventional end-to-end models require substantially more training iterations to achieve their peak accuracy. This rapid convergence is a direct result of our Predictive-Corrective paradigm, which creates a smoother and more tractable optimization landscape.

Table 2: **Ablation study on the OASIS-1 and MRBrainS13 test sets, validating the contributions of our core components.** The results demonstrate that removing either the **Predictive Prior Module (PPM)** or the **Corrective Residual Network (CRN)** leads to a degradation in performance. Our full model (**PCMambaNet**, highlighted), which integrates both components, consistently achieves the best overall results (in **bold**), confirming their essential roles and synergistic effect. Dice and IOU are reported as percentages, while HD95 and ASD are scaled by a factor of 10 for readability.

| Configuration | Dice (%) ↑ | | | HD95 (mm×10)↓ | | | ASD (mm×10)↓ | | | IOU (%)↑ | | |
|---|---|---|---|---|---|---|---|---|---|---|---|---|
| | CSF | GM | WM | CSF | GM | WM | CSF | GM | WM | CSF | GM | WM |
| **OASIS-1 dataset** | | | | | | | | | | | | |
| PCMambaNet (Full Model) | **94.10 ± 0.38** | **94.33 ± 0.35** | **94.29 ± 0.34** | **10.58 ± 0.03** | 10.81 ± 0.17 | 15.00 ± 1.25 | **1.76 ± 0.02** | 1.88 ± 0.18 | 2.70 ± 0.38 | **89.91 ± 0.08** | **91.22 ± 0.50** | **92.23 ± 0.66** |
| (1) CRN only (w/o PPM) | 92.29 ± 1.35 | 93.22 ± 0.99 | 93.87 ± 1.22 | 13.66 ± 2.44 | 12.36 ± 2.15 | 20.92 ± 7.66 | 7.10 ± 0.96 | 2.80 ± 0.12 | 7.01 ± 0.95 | 87.37 ± 2.32 | 89.24 ± 1.73 | 89.43 ± 2.00 |
| (2) w/ Simple PPM (Random Mask) | 93.53 ± 0.18 | 94.22 ± 0.20 | 94.21 ± 0.67 | 10.65 ± 0.01 | **10.74 ± 0.04** | **14.96 ± 0.33** | 1.91 ± 0.10 | **1.83 ± 0.06** | **2.48 ± 0.09** | 89.35 ± 0.50 | 90.86 ± 0.75 | 91.74 ± 0.47 |
| (3) PPM only (w/o CRN) | 91.23 ± 1.58 | 93.46 ± 0.37 | 93.49 ± 0.41 | 19.83 ± 0.00 | 11.65 ± 0.32 | 18.19 ± 0.69 | 9.38 ± 0.65 | 2.51 ± 0.12 | 3.69 ± 0.63 | 82.42 ± 2.96 | 88.88 ± 0.61 | 89.74 ± 0.66 |
| (4) w/ CNN-CRN | 91.52 ± 2.31 | 93.35 ± 0.49 | 93.58 ± 0.61 | 13.91 ± 4.58 | 11.57 ± 0.46 | 17.00 ± 0.90 | 3.38 ± 1.46 | 2.48 ± 0.39 | 2.90 ± 0.43 | 86.65 ± 4.40 | 89.70 ± 0.93 | 90.42 ± 0.64 |
| **MRBrainS13 dataset** | | | | | | | | | | | | |
| PCMambaNet (Full Model) | **69.37 ± 0.41** | **71.79 ± 0.80** | **74.26 ± 0.93** | **19.31 ± 1.98** | **17.46 ± 2.00** | **38.90 ± 7.33** | **4.47 ± 0.23** | 4.60 ± 0.51 | 14.41 ± 2.18 | **60.67 ± 0.97** | **64.63 ± 1.14** | **67.48 ± 1.08** |
| (1) CRN only (w/o PPM) | 67.40 ± 1.90 | 70.16 ± 1.65 | 72.53 ± 1.87 | 21.29 ± 2.47 | 27.68 ± 0.94 | 39.50 ± 0.69 | 5.49 ± 0.95 | 7.92 ± 3.17 | 19.96 ± 4.49 | 58.06 ± 2.73 | 62.18 ± 2.46 | 64.88 ± 2.60 |
| (2) w/ Simple PPM (Random Mask) | 65.71 ± 0.45 | 68.67 ± 0.08 | 71.16 ± 0.26 | 24.94 ± 0.53 | 27.38 ± 1.26 | 52.87 ± 2.28 | 6.02 ± 0.09 | 7.04 ± 0.46 | 21.62 ± 3.01 | 55.72 ± 0.60 | 60.28 ± 0.06 | 63.09 ± 0.37 |
| (3) PPM only (w/o CRN) | 68.84 ± 1.64 | 70.13 ± 1.74 | 73.29 ± 2.05 | 22.29 ± 2.22 | 27.55 ± 2.45 | 52.48 ± 1.73 | 5.61 ± 0.70 | 7.01 ± 2.99 | 21.07 ± 5.78 | 57.29 ± 2.33 | 61.04 ± 2.38 | 64.23 ± 2.68 |
| (4) w/ CNN-CRN | 66.58 ± 2.38 | 69.57 ± 2.00 | 71.58 ± 2.43 | 25.46 ± 5.35 | 25.79 ± 2.86 | 48.71 ± 2.70 | 5.96 ± 1.31 | 6.76 ± 2.32 | 17.64 ± 4.81 | 56.99 ± 3.22 | 61.52 ± 2.79 | 63.95 ± 3.25 |

paradigm; this conclusion is further supported by the additional quantitative results in Appendix A.4 (Table 6).

## 3.5 DATA EFFICIENCY ANALYSIS

Our **PCMambaNet** exhibits strong data efficiency, achieving near-saturated segmentation performance with only a small amount of labeled data. As shown in Table 3, when trained on merely **10%** of the OASIS-1 dataset, PCMambaNet attains the highest Dice scores for CSF/GM/WM, not only outperforming all competing methods under the same data budget, but also surpassing the best GM Dice (92.89%) achieved by any baseline trained on the full **100%** dataset; for WM, the Dice score of 92.86% at 10% data is also very close to the fully supervised baselines (up to 93.31%). As the labeled fraction increases from **10%** to **25%**, **50%**, and **100%**, PCMambaNet consistently achieves the best Dice, HD95, ASD, and IOU scores across all three tissue classes, while the marginal performance gains remain relatively small (e.g., GM Dice improves only from **93.11%** to **94.33%**). This behavior indicates that PCMambaNet can already learn robust and well-regularized representations under limited supervision, with additional data mainly providing fine-grained refinement. We attribute this pronounced "capital efficiency" (i.e., data efficiency) to the PC paradigm itself: by explicitly injecting domain priors into the architecture, PCMambaNet substantially reduces its reliance on large-scale annotated datasets, making it particularly suitable for medical imaging scenarios where labeled data are scarce; further quantitative evidence is provided in Appendix A.4 (Table 7).

Table 3: **PCMambaNet demonstrates superior data efficiency on the OASIS-1 dataset.** Our model consistently outperforms the baseline when trained on fractions of the data. Notably, **PC-MambaNet** using just [e.g., 10%] of the data surpasses the baseline trained on the entire dataset. Dice and IOU are reported as percentages, while HD95 and ASD are scaled by a factor of 10 for readability.

| Number | Model | Dice (%) ↑ | | | HD95 (mm×10)↓ | | | ASD (mm×10)↓ | | | IOU (%)↑ | | |
|---|---|---|---|---|---|---|---|---|---|---|---|---|---|
| | | CSF | GM | WM | CSF | GM | WM | CSF | GM | WM | CSF | GM | WM |
| 10% | UNet | 85.12 | 87.77 | 87.72 | 19.22 | 20.84 | 44.54 | 4.31 | 6.19 | 15.89 | 75.70 | 80.33 | 80.86 |
| | nnUNet | 87.22 | 89.46 | 89.25 | 18.46 | 16.54 | 30.96 | 5.59 | 4.94 | 9.57 | 78.89 | 83.29 | 83.11 |
| | Swin-UNet | 90.21 | 91.73 | 91.90 | 13.99 | 13.45 | 22.55 | 4.08 | 3.74 | 5.88 | 83.70 | 86.71 | 87.07 |
| | Mamba-UNet | 88.34 | 91.55 | 91.43 | 18.62 | 13.25 | 21.67 | 5.28 | 3.62 | 5.17 | 80.64 | 86.23 | 86.35 |
| | PCMambaNet | **92.79** | **93.11** | **92.86** | **10.83** | **11.73** | **19.22** | **2.21** | **2.59** | **3.80** | **88.15** | **88.96** | **88.80** |
| 25% | UNet | 86.31 | 87.40 | 87.52 | 18.31 | 19.75 | 42.69 | 4.09 | 6.04 | 15.28 | 77.51 | 79.67 | 80.26 |
| | nnUNet | 87.84 | 90.33 | 90.70 | 18.51 | 14.40 | 24.51 | 5.57 | 4.07 | 7.23 | 79.86 | 84.67 | 85.27 |
| | Swin-UNet | 90.62 | 92.02 | 92.04 | 14.26 | 12.91 | 21.14 | 4.15 | 3.49 | 4.93 | 84.38 | 87.12 | 87.40 |
| | Mamba-UNet | 89.07 | 91.73 | 92.03 | 18.76 | 12.91 | 20.85 | 5.26 | 3.31 | 4.73 | 80.26 | 86.58 | 87.26 |
| | PCMambaNet | **92.90** | **93.31** | **93.47** | **10.68** | **11.45** | **18.02** | **2.06** | **2.33** | **3.65** | **88.49** | **89.49** | **89.75** |
| 50% | UNet | 88.62 | 90.27 | 91.25 | 19.54 | 15.00 | 24.01 | 4.47 | 4.04 | 7.08 | 77.95 | 84.11 | 86.03 |
| | nnUNet | 87.98 | 90.94 | 91.42 | 18.83 | 13.18 | 21.26 | 5.60 | 3.60 | 5.95 | 80.10 | 85.56 | 86.39 |
| | Swin-UNet | 91.00 | 92.26 | 92.57 | 13.21 | 12.41 | 20.49 | 3.73 | 3.25 | 4.78 | 85.15 | 87.62 | 88.20 |
| | Mamba-UNet | 88.92 | 91.99 | 92.42 | 17.31 | 12.61 | 20.51 | 4.68 | 3.31 | 4.72 | 81.64 | 87.03 | 87.88 |
| | PCMambaNet | **92.90** | **93.63** | **93.79** | **11.02** | **11.24** | **17.18** | **2.16** | **2.16** | **3.10** | **88.45** | **89.98** | **90.29** |
| 100% | UNet | 91.18 | 92.43 | 93.16 | 12.54 | 12.67 | 19.66 | 2.69 | 2.95 | 5.20 | 85.43 | 87.91 | 89.20 |
| | nnUNet | 90.38 | 92.09 | 92.92 | 14.17 | 11.63 | 17.77 | 4.25 | 3.07 | 5.12 | 83.99 | 87.38 | 88.65 |
| | Swin-UNet | 90.98 | 92.43 | 92.57 | 13.53 | 12.02 | 19.44 | 3.88 | 3.08 | 4.82 | 85.05 | 87.92 | 88.76 |
| | Mamba-UNet | 91.95 | 92.89 | 93.31 | 11.60 | 11.78 | 18.75 | 2.74 | 2.76 | 4.26 | 86.73 | 88.77 | 89.25 |
| | PCMambaNet | **94.10** | **94.33** | **94.29** | **10.58** | **10.81** | **15.00** | **1.76** | **1.88** | **2.70** | **89.91** | **91.22** | **92.23** |

## 3.6 ANALYSIS OF INTERNAL FEATURE REPRESENTATIONS

To better understand how PCMambaNet operates under the proposed PC paradigm, we visualize its internal feature maps and compare them with those of Mamba-UNet in Figure 5. After only **5 epochs** of training, PCMambaNet already exhibits more focused and interpretable representations. At shallow and middle layers, its feature maps concentrate on salient brain tissues and suppress many spurious background responses, indicating that the anatomy-informed **PPM** effectively guides attention toward relevant regions from the early stage of training. At deeper layers, the effect of the **CRN** becomes apparent: PCMambaNet captures finer textures and more coherent structural boundaries, while the baseline features remain comparatively diffuse. Overall, these visualizations suggest that the predictive–corrective design helps allocate model capacity to diagnostically meaningful structures and filter out redundant activations, contributing to more efficient use of training data.

## 3.7 ANALYSIS OF CLASS ACTIVATION MAPS

To further investigate the model's decision-making process, we utilize Class Activation Maps (CAM) to visualize class-specific attention, with results presented in Figure 6. The heatmaps reveal that **PCMambaNet** learns to accurately localize target tissues for each class with remarkable efficiency. After only **5 epochs** on limited data, our model generates clean, well-defined activation maps that focus precisely on the relevant anatomical structures. While the baseline Mamba-UNet's performance improves with more data (e.g., 200 epochs), its activation maps often remain diffuse or highlight irrelevant regions. In stark contrast, **PCMambaNet**, with just **5 epochs** on the same large-scale dataset, produces significantly more focused and semantically meaningful heatmaps. This superior localization ability provides strong evidence for our central thesis: the **PPM** effectively guides the model's focus, while the **CRN** refines the representation, enabling the network to rapidly learn and generalize the intrinsic characteristics of each tissue class.

## 4 LIMITATIONS

Despite the promising results of the PC paradigm, several limitations remain. First, its effectiveness relies on well-defined domain priors that can be encoded into the PPM, requiring task-specific design. This makes the paradigm less "plug-and-play" than fully generic end-to-end models, particularly for problems lacking clear structural priors. Second, the framework may be sensitive to error propagation: if the PPM produces inaccurate initial predictions, the CRN's corrective ability can be constrained. Finally, because the overall architecture is built around anatomy-informed priors, there

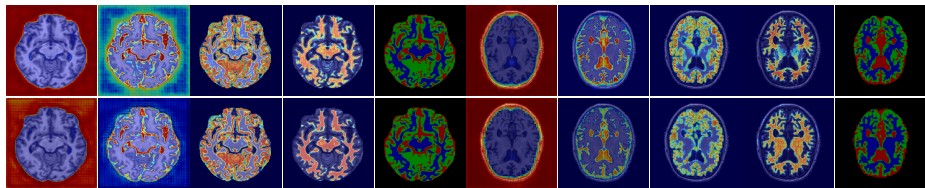

Figure 5: The visualization of internal feature maps illustrates the effectiveness of our Predictive–Corrective paradigm. The **PCMambaNet**, trained for only **5 epochs** (bottom rows), shows more focused responses to salient anatomical structures and better suppression of background noise across shallow (first encoder layer), middle (bottleneck layer), and deep (third decoder layer) features. In contrast, the representations extracted by Mamba-UNet (top rows) remain relatively indistinct even after 50 epochs of training. Comparative results are shown on the OASIS-1 (left) and MRBrainS13 (right) datasets.

Figure 6: Qualitative comparison demonstrating the efficient learning behavior and high-quality segmentation results of **PCMambaNet**. Despite being trained for only **5 epochs**, our **PCMambaNet** (bottom rows) produces significantly more accurate and well-defined segmentation heatmaps on both the OASIS-1 and MRBrainS13. This performance markedly surpasses the baseline Mamba-UNet (top rows), which was trained for 50/200 epochs. For each dataset, columns display heatmaps for individual classes (Background, CSF, GM, WM) followed by the final prediction overlay.

is a certain computational overhead in terms of parameter count, GFLOPs, throughput, and inference time, as shown in Table 4. These factors indicate potential directions for future improvements in balancing structural priors with computational efficiency.

Table 4: Comparison of model parameters, FLOPs, throughput, and inference time (batch size = 1).

| Model | Params (M) ↓ | GFLOPs ↓ | Throughput (FPS) ↑ | Inference time (ms/batch) ↓ |
|---|---|---|---|---|
| U-Net | **1.81** | **2.28** | **267.00** | **2.20** |
| nnNet | 18.69 | 3.25 | 127.01 | 4.94 |
| Swin-UNet | 41.38 | 8.98 | 53.86 | 8.92 |
| Mamba-UNet | 35.86 | 7.65 | 37.20 | 11.86 |
| **PCMambaNet (Ours)** | 92.88 | 20.12 | 20.61 | 32.12 |

## 5 CONCLUSION

In this paper, we challenged the dominant end-to-end learning paradigm in deep learning, highlighting its limitations in training efficiency and data dependence, particularly in data-scarce domains such as medical imaging. To address this bottleneck, we proposed the **Predictive–Corrective (PC) paradigm**, which decomposes a complex segmentation task into a lightweight prior prediction stage (PPM) and a powerful residual correction stage (CRN). On brain MRI segmentation, our PCMambaNet instantiation achieves competitive, and in some cases state-of-the-art, accuracy with relatively few training epochs, and extensive ablation and data-efficiency analyses confirm that the complementary roles of PPM and CRN are key to these improvements and to the robustness of the learned representations under limited data. Nevertheless, the current PC instantiation still relies on manually designed priors and has so far been validated only on brain MRI segmentation, pointing to future work on more automated prior construction and broader applications across tasks and imaging modalities.

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

# Appendix to The Predictive-Corrective Paradigm: Decoupling Prediction and Refinement for Efficient Anatomy-Informed Brain MRI Segmentation

In this appendix, we provide the following materials:

A. Supplementary details in the main paper;

C. Hyperparameter $\theta$ Analysis;

D. Proof of the method;

E. LLM Contribution.

## A  SUPPLEMENTARY DETAILS

### A.1  RELATED WORK

In recent years, with the continuous advancement of deep learning, medical image segmentation techniques have undergone rapid updates and iterations (Rayed et al., 2024; Azad et al., 2024). In the early stages of this field, convolutional neural network (CNN)-based methods were widely adopted for various segmentation tasks (Azad et al., 2024; Ronneberger et al., 2015). The fully convolutional network (FCN) pioneered end-to-end pixel-wise segmentation (Long et al., 2015). UNet (Ronneberger et al., 2015) and its variants (such as UNet++ (Zhou et al., 2018), UNet3+ (Huang et al., 2020), and nnUNet (Isensee et al., 2021)) have significantly improved the fusion of multi-scale features and the localization of lesion boundaries through a symmetric encoder-decoder structure with multi-level skip connections (Li et al., 2023). However, due to the limited local receptive field of CNN-based models, it remains challenging for them to effectively capture global contextual information, resulting in performance bottlenecks for medical image segmentation tasks involving complex structures or long-range dependencies (Dosovitskiy et al., 2020; Gu & Dao, 2023; Zhu et al., 2024; Ren et al., 2022; 2024a;b).

To enhance the modeling capability of global contextual information, Transformers (Vaswani et al., 2017) and their vision variants (ViT) (Dosovitskiy et al., 2020) have gradually become mainstream approaches for medical image segmentation. Methods such as TransUNet (Chen et al., 2021), UT-Net (Gao et al., 2021), ViT-UNet (Zhou et al., 2024), and Swin-UNet (Cao et al., 2022) incorporate self-attention mechanisms to efficiently integrate multi-level features, thereby substantially improving segmentation accuracy. Further, models like DS-TransUNet (Lin et al., 2022) and TransFuse (Zhang et al., 2021) explore the integration of parallel architectures and multi-branch designs, enabling effective fusion of both local and global information. However, Transformer-based models suffer from quadratic computational complexity concerning self-attention (Gu & Dao, 2023), which poses significant challenges in terms of efficiency and resource consumption when processing high-resolution medical images or deploying on edge devices (Zhang et al., 2024a; Zhu et al., 2024; Guo et al., 2024; Chen et al., 2024a; Diao et al., 2025).

In recent years, state space models (SSMs) represented by Mamba have emerged as a promising research direction in medical image segmentation, owing to their linear computational complexity and strong capability for long-range dependency modeling. VMamba (Liu et al., 2024b) first introduced a multi-directional scanning vision Mamba backbone network, demonstrating excellent performance in medical image segmentation tasks (Ruan et al., 2024). Building upon Swin-UNet (Cao et al., 2022), Mamba-UNet (Wang et al., 2024b) incorporated pure vision Mamba modules into the segmentation architecture, significantly improving both segmentation accuracy and inference efficiency. To meet practical deployment requirements, lightweight vision Mamba variants such as LightM-UNet (Liao et al., 2024) and UltraLight VM-UNet (Wu et al., 2024) were proposed, offering extremely low parameter counts and high computational efficiency suitable for mobile and resource-constrained environments. MedMamba (Yue & Li, 2024) systematically applied Mamba to multi-modal medical image classification and segmentation, utilizing a hybrid structure that combines convolutional layers with state space modeling to balance local detail and global context. To further enhance spatial structural awareness and multi-scale information fusion, many Mamba-based

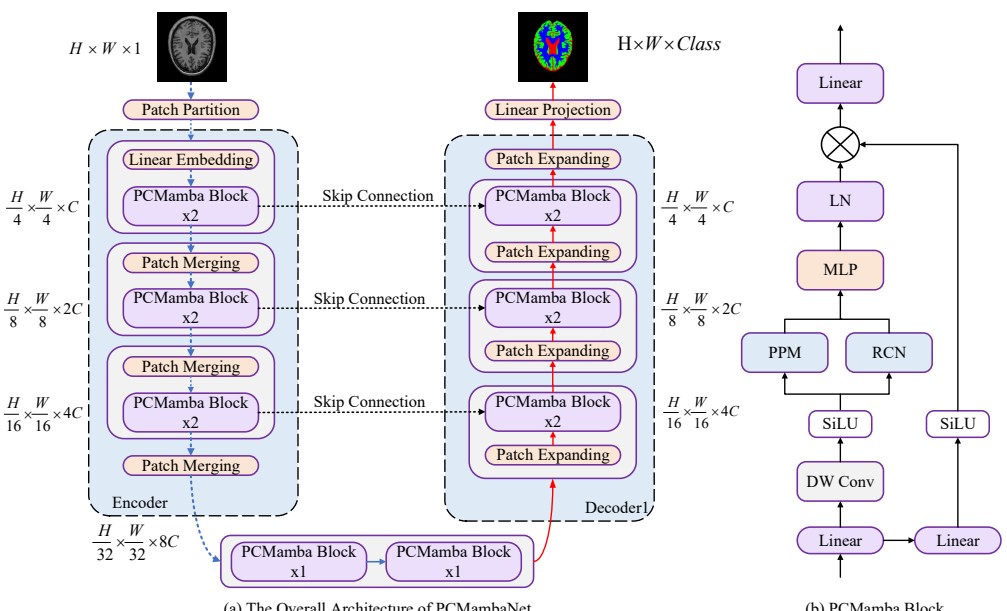

Figure 7: (a) The overall architecture of our proposed PCMambaNet, which follows a U-Net–like structure with our novel PCMamba Blocks as the core building components. (b) The structural diagram of the PCMamba Block.

variants have introduced various architectural innovations. For instance, LMa-UNet (Wang et al., 2024a) inserts large-window state space modules at multiple scales in the UNet encoder to expand the receptive field and improve global modeling capability. LocalMamba ()localmamba employs window-based local scanning and dynamic directional search to strengthen local spatial correlations effectively. Selective and Multi-Scale Fusion Mamba (SMM-UNet) (Li et al., 2025) proposes multi-scale feature fusion and selective dynamic weighting mechanisms, achieving precise segmentation of complex lesion structures with a minimal number of parameters. Spatial-Mamba (Xiao et al., 2025) introduces a structure-aware state fusion module to directly aggregate local and global information in the latent space, further enhancing the modeling of complex spatial structures and ambiguous boundaries.

In addition, domain generalization and three-dimensional medical image segmentation represent important application directions for Mamba-based architectures (Xie et al.). Approaches such as Mamba-Sea (Cheng et al., 2025) and SegMamba (Xing et al., 2024) integrate global-local sequence enhancement with 3D state space modeling, thereby improving the generalization ability and spatial consistency of segmentation across multi-center and multi-modal medical datasets. VMAXL-UNet (Zhong et al., 2025) achieves both high accuracy and efficiency on multiple medical segmentation benchmarks by combining vision Mamba modules with lightweight LSTM (Greff et al., 2016) components.

Overall, Mamba and its variants have achieved multidimensional breakthroughs in medical image segmentation, ranging from lightweight design and structure awareness to multi-scale fusion, three-dimensional modeling, and domain generalization. These advances have extensively promoted the development of efficient, accurate, and generalizable segmentation models. However, most existing Mamba-based models rely on end-to-end optimization; thus, their convergence speed and generalization ability still have room for improvement, especially in real-world clinical scenarios involving limited data and the incorporation of prior knowledge. To address these challenges, this paper proposes a prediction-correction paradigm that integrates domain knowledge with an efficient Mamba architecture, aiming to achieve a more efficient and robust solution for medical image segmentation.

## A.2 ARCHITECTURE OVERVIEW

Figure 7 (a) shows the segmentation architecture of the proposed PCMambaNet. First, the input 2D image of size $H \times W \times 1$ is divided into patches and then flattened into a one-dimensional se-

quence. This sequence is then projected to a dimension of $C$ through a linear embedding layer and processed through a series of Predictive-Corrective Mamba (PCMamba) Blocks and downsampling layers. Each encoder stage in PCMambaNet extracts features using two PCMamba blocks, and the feature map sizes at each stage are $H/4 \times W/4$, $H/8 \times W/8$, $H/16 \times W/16$, and $H/32 \times W/32$, respectively. The bottleneck of PCMambaNet is composed of two PCMamba Blocks. Symmetrically, each decoder stage also utilizes two PCMamba Blocks for feature reconstruction, with feature map sizes of $H/16 \times W/16$, $H/8 \times W/8$, and $H/4 \times W/4$, respectively. Skip connections are used to fuse multi-scale features from the encoder to the decoder.

### A.3 FUSION AND STATE MODULATION

In the final step, the outputs from the predictive branch ($z_i^{\text{mask}}$) and the corrective branch ($z_i^{\text{density}}$) are concatenated and fused to generate the final contextual modulation factor, $z_i^{\text{fused}}$. We employ a lightweight MLP for this fusion, as it can learn the optimal non-linear combination of the two signals, which is more powerful than a simple linear aggregation:

$$z_i^{\text{fused}} = \text{MLP}\left([z_i^{\text{mask}}; z_i^{\text{density}}]\right). \tag{10}$$

This fused factor is then used to modulate the original Mamba state $x_t$ as shown in Eq. 6. By synergistically integrating the "where to look" guidance from the PPM with the "how to refine" details from the CRN, the resulting state $h_t$ becomes significantly more informed.

### A.4 QUANTITATIVE

To provide a more comprehensive and granular validation of our model, we present supplementary results across several standard evaluation metrics. The detailed metrics in Table 5 further substantiate the superiority of PCMambaNet, demonstrating its consistent outperformance against all baseline models across Accuracy (Acc), Precision (Pre), Sensitivity (Sen), and Specificity (Spe) on both the OASIS-1 and MRBrainS13 datasets. Similarly, the supplementary ablation results in Table 6 reinforce our core findings; these metrics confirm that the removal or simplification of either the Predictive Prior Module (PPM) or the Residual Corrective Network (CRN) leads to a general degradation in performance, underscoring their individual necessity and synergistic effect. Finally, Table 7 offers a more detailed view of our model's remarkable data efficiency, showing strong performance across all metrics even when trained with only a fraction of the data. Collectively, these results provide robust, multi-faceted evidence for the effectiveness and efficiency of the proposed PC paradigm.

Table 5: Quantitative comparison on the OASIS-1 and MRBrainS13 test sets. Our method (highlighted with a light background) achieves accuracy comparable to state-of-the-art (SOTA) approaches while exhibiting significant advantages across all efficiency metrics. All metrics (Acc, Pre, Sen, Spe) are reported as percentages, and the best results are highlighted in bold.

| Model | Acc (%)↑ | | | Pre (%)↑ | | | Sen (%)↑ | | | Spe (%)↑ | | |
|---|---|---|---|---|---|---|---|---|---|---|---|---|
| | CSF | GM | WM | CSF | GM | WM | CSF | GM | WM | CSF | GM | WM |
| **OASIS-1 dataset** | | | | | | | | | | | | |
| UNet | 97.44±0.06 | 96.89±0.15 | 98.71±0.07 | **94.37±0.95** | 92.33±0.79 | 91.91±1.08 | 88.59±1.98 | 92.78±0.47 | **95.91±0.60** | 98.08±0.74 | **97.97±0.71** | 98.06±0.80 |
| nnUNet | 97.37±0.01 | 96.86±0.01 | 98.66±0.01 | 88.80±0.04 | 92.36±0.11 | 92.91±0.06 | 92.30±0.04 | 91.96±0.05 | 94.45±0.06 | 97.69±0.01 | 97.40±0.01 | 98.91±0.01 |
| Swin-UNet | 97.42±0.02 | 96.92±0.03 | 98.70±0.02 | 90.44±0.36 | 92.69±0.18 | 94.06±0.17 | 92.38±0.07 | 92.66±0.13 | 93.24±0.47 | 97.74±0.02 | 97.37±0.03 | 99.00±0.01 |
| Mamba-UNet | 97.44±0.05 | 96.94±0.01 | 98.73±0.04 | 92.66±0.12 | 92.71±0.62 | 94.04±0.18 | 91.63±0.45 | 93.51±0.27 | 93.99±0.44 | 97.86±0.03 | 97.30±0.01 | 98.98±0.02 |
| PCMambaNet(1 epoch) | 96.57±0.01 | 95.84±0.22 | 97.19±0.03 | 91.32±0.02 | 90.88±0.23 | 92.45±0.33 | 89.97±2.51 | 91.88±0.33 | 91.66±0.22 | 97.77±0.05 | 96.28±0.04 | 98.24±0.03 |
| PCMambaNet(5 epochs) | 97.13±0.01 | 96.57±0.04 | 97.91±0.02 | 92.59±0.02 | 91.79±0.01 | 94.37±0.04 | 91.55±0.21 | 93.16±0.30 | 92.88±0.18 | 96.81±0.01 | 96.11±0.06 | 98.51±0.05 |
| PCMambaNet(50 epochs) | **97.56±0.04** | **97.12±0.08** | **98.77±0.06** | 93.69±0.30 | **93.42±1.75** | **94.79±0.11** | **93.33±0.21** | **95.02±0.51** | 93.79±0.11 | **98.81±0.06** | 97.36±0.21 | **99.12±0.04** |
| **MRBrainS13 dataset** | | | | | | | | | | | | |
| UNet | 79.18±0.66 | 78.98±0.66 | 89.59±1.76 | **71.27±0.73** | **70.59±0.13** | 73.30±0.36 | 65.02±0.78 | 70.81±0.64 | **77.59±0.91** | 80.25±0.61 | 79.84±0.59 | 89.82±1.76 |
| nnUNet | 78.52±0.27 | 78.42±0.26 | **89.95±4.08** | 62.34±1.22 | 65.15±1.98 | 64.75±1.01 | 62.28±2.59 | 68.07±0.85 | 69.04±1.50 | 79.83±0.06 | 79.55±0.12 | **90.41±4.18** |
| Swin-UNet | 78.99±0.25 | 78.28±0.72 | 88.45±2.78 | 65.83±1.58 | 67.66±1.58 | 71.18±1.96 | 64.10±1.80 | 70.59±0.34 | 71.57±1.20 | 80.11±0.13 | 79.07±0.70 | 88.95±2.77 |
| Mamba-UNet | 79.08±0.03 | 79.03±0.07 | 88.80±1.15 | 67.03±0.04 | 67.75±0.50 | 70.10±1.27 | 63.61±0.72 | 70.67±0.40 | 74.19±0.62 | 80.23±0.02 | 79.84±0.05 | 89.23±1.16 |
| PCMambaNet(1 epoch) | 77.19±0.06 | 76.52±0.09 | 76.99±0.23 | 59.88±0.14 | 56.39±0.22 | 65.89±0.31 | 56.59±0.07 | 65.88±0.12 | 46.87±0.23 | 78.74±0.02 | 77.85±0.31 | 78.54±0.41 |
| PCMambaNet(5 epochs) | 77.44±0.52 | 77.77±0.29 | 78.72±0.02 | 62.89±0.33 | 65.79±0.22 | 64.64±3.58 | 63.92±0.42 | 66.89±0.65 | 66.96±0.41 | 78.95±0.04 | 78.75±0.03 | 78.43±0.88 |
| PCMambaNet(200 epochs) | **79.62±0.11** | **79.49±0.10** | 89.72±0.09 | 70.73±0.67 | 70.42±0.82 | **74.50±2.42** | **68.45±0.12** | **73.81±0.35** | 74.71±0.18 | **80.49±0.05** | **80.11±0.08** | 90.18±0.04 |

Table 6: Ablation study results on the OASIS-1 and MRBrainS13 test sets, where the full model is highlighted with a light background. All metrics (Acc, Pre, Sen, Spe) are reported as percentages, and the best-performing results are highlighted in bold.

| Configuration | Acc (%)↑ | | | Pre (%)↑ | | | Sen (%)↑ | | | Spe (%)↑ | | |
|---|---|---|---|---|---|---|---|---|---|---|---|---|
| | CSF | GM | WM | CSF | GM | WM | CSF | GM | WM | CSF | GM | WM |
| **OASIS-1 dataset** | | | | | | | | | | | | |
| PCMambaNet (Full Model) | **97.56 ± 0.04** | 97.12 ± 0.08 | **98.77 ± 0.06** | 93.69 ± 0.30 | 93.42 ± 1.75 | 94.79 ± 0.11 | **93.33 ± 0.21** | **95.02 ± 0.51** | 93.79 ± 0.11 | **98.81 ± 0.06** | **97.36 ± 0.21** | **99.12 ± 0.04** |
| (1) CRN only (w/o PPM) | 97.54 ± 0.10 | 97.04 ± 0.19 | 98.67 ± 0.04 | **93.89 ± 0.47** | 92.59 ± 0.87 | 93.96 ± 2.88 | 91.03 ± 2.09 | 94.11 ± 1.05 | 94.07 ± 0.97 | 97.95 ± 0.02 | 97.29 ± 0.12 | 98.84 ± 0.05 |
| (2) w/ Simple PPM (Random Mask) | 97.53 ± 0.04 | **97.15 ± 0.07** | 98.16 ± 0.58 | 93.80 ± 0.52 | 93.41 ± 0.28 | **96.04 ± 0.64** | 93.15 ± 0.23 | 94.82 ± 0.64 | **94.16 ± 0.17** | 97.81 ± 0.05 | 97.33 ± 0.06 | 98.74 ± 0.56 |
| (3) PPM only (w/o CRN) | 97.21 ± 0.31 | 96.97 ± 0.13 | 98.74 ± 0.08 | 88.22 ± 2.06 | 92.42 ± 0.91 | 94.94 ± 0.49 | 90.78 ± 1.41 | 93.86 ± 0.39 | 93.38 ± 1.77 | 97.60 ± 0.22 | 97.26 ± 0.20 | 99.01 ± 0.05 |
| (4) w/ CNN-CRN | 97.43 ± 0.27 | 97.03 ± 0.14 | 98.46 ± 0.51 | 91.49 ± 3.11 | 92.94 ± 0.77 | 95.41 ± 0.40 | 92.11 ± 1.65 | 94.25 ± 0.29 | 93.30 ± 0.38 | 97.77 ± 0.19 | 97.32 ± 0.14 | 99.03 ± 0.07 |
| **MRBrainS13 dataset** | | | | | | | | | | | | |
| PCMambaNet (Full Model) | **79.62 ± 0.11** | **79.49 ± 0.10** | 89.72 ± 0.09 | **70.73 ± 0.67** | **70.42 ± 0.82** | 74.50 ± 2.42 | **68.45 ± 0.12** | **73.81 ± 0.35** | **74.71 ± 0.18** | **80.49 ± 0.05** | **80.11 ± 0.08** | 90.18 ± 0.04 |
| (1) CRN only (w/o PPM) | 79.36 ± 0.26 | 79.19 ± 0.30 | 90.30 ± 2.26 | 70.16 ± 2.12 | 68.76 ± 1.98 | 73.84 ± 1.65 | 67.00 ± 1.68 | 71.96 ± 1.18 | 73.01 ± 1.44 | 80.28 ± 0.16 | 79.92 ± 0.21 | 90.78 ± 2.32 |
| (2) w/ Simple PPM (Random Mask) | 79.13 ± 0.04 | 79.01 ± 0.03 | 90.21 ± 0.58 | 66.18 ± 0.33 | 68.51 ± 0.12 | 72.22 ± 1.18 | 65.61 ± 1.11 | 69.49 ± 0.13 | 71.12 ± 0.73 | 80.04 ± 0.06 | 79.89 ± 0.02 | 90.70 ± 0.56 |
| (3) PPM only (w/o CRN) | 79.29 ± 0.22 | 79.13 ± 0.27 | **90.58 ± 2.61** | 67.53 ± 1.66 | 68.62 ± 2.04 | 72.45 ± 1.52 | 66.54 ± 1.59 | 70.07 ± 1.30 | 74.26 ± 1.68 | 80.24 ± 0.11 | 79.97 ± 0.22 | **91.03 ± 2.67** |
| (4) w/ CNN-CRN | 79.28 ± 0.31 | 79.14 ± 0.32 | 89.90 ± 2.00 | 68.09 ± 2.29 | 68.71 ± 2.02 | 72.28 ± 1.76 | 65.63 ± 2.30 | 70.88 ± 1.76 | 74.00 ± 2.00 | 80.29 ± 0.15 | 79.92 ± 0.22 | 90.35 ± 2.10 |

Table 7: Data efficiency comparison on the OASIS-1 dataset, where the metrics highlighted with a light background correspond to models trained with the entire dataset. All metrics (Acc, Pre, Sen, Spe) are reported as percentages.

| Number | Model | Acc (%)↑ | | | Pre (%)↑ | | | Sen (%)↑ | | | Spe (%)↑ | | |
|---|---|---|---|---|---|---|---|---|---|---|---|---|---|
| | | CSF | GM | WM | CSF | GM | WM | CSF | GM | WM | CSF | GM | WM |
| 10% | UNet | 96.85 | 95.85 | 98.12 | 92.05 | 87.45 | 82.86 | 79.72 | 88.56 | **96.57** | 97.90 | 96.71 | 98.13 |
| | nnUNet | 97.02 | 96.31 | 98.35 | 86.48 | 89.27 | 88.28 | 88.88 | 90.13 | 93.01 | 97.61 | 97.06 | 98.59 |
| | Swin-UNet | 97.30 | 96.70 | 98.56 | 90.52 | 91.13 | 92.71 | 90.37 | 92.59 | 93.08 | 97.75 | 97.16 | 98.89 |
| | Mamba-UNet | 97.07 | 96.57 | 98.49 | 87.99 | 90.47 | 93.47 | 89.22 | 92.98 | 91.42 | 97.60 | 97.00 | 98.88 |
| | PCMambaNet | **97.58** | **97.00** | **98.70** | **92.98** | **93.08** | **94.55** | **92.93** | **93.37** | 93.03 | 97.89 | **97.40** | **99.00** |
| 25% | UNet | 96.96 | 95.71 | 97.89 | 92.75 | 88.44 | 81.68 | 81.18 | 86.81 | 71.21 | **97.94** | 96.91 | 97.76 |
| | nnUNet | 97.07 | 96.50 | 98.46 | 86.26 | 90.37 | 90.46 | 90.19 | 90.64 | **93.34** | 97.56 | 97.15 | 98.75 |
| | Swin-UNet | 97.33 | 96.75 | 98.58 | 89.58 | 92.37 | 93.09 | 92.17 | 91.93 | 92.92 | 97.68 | **97.38** | 98.82 |
| | Mamba-UNet | 97.05 | 96.64 | 98.58 | 87.90 | 90.86 | 93.41 | 88.79 | 92.95 | 92.48 | 97.60 | 97.03 | 98.90 |
| | PCMambaNet | **97.46** | **96.96** | **98.77** | **93.60** | 92.84 | **95.35** | 92.49 | **94.06** | 93.21 | 97.78 | 97.20 | **99.10** |
| 50% | UNet | 96.94 | 96.30 | 98.42 | 91.01 | 89.76 | 88.76 | 83.02 | 91.09 | **95.78** | 97.84 | 96.91 | 98.46 |
| | nnUNet | 97.09 | 96.61 | 98.53 | 86.14 | 90.63 | 91.99 | 90.55 | 91.63 | 93.07 | 97.54 | 97.15 | 98.85 |
| | Swin-UNet | 97.40 | 96.85 | 98.66 | 90.74 | 92.40 | 93.81 | 91.80 | 92.44 | 93.08 | 97.76 | **97.31** | 98.98 |
| | Mamba-UNet | 97.15 | 96.72 | 98.62 | 89.03 | 91.46 | 92.93 | 89.33 | 92.86 | 93.62 | 97.66 | 97.14 | 99.90 |
| | PCMambaNet | **97.52** | **97.07** | **98.80** | **93.67** | **93.02** | **95.85** | 92.40 | **94.47** | 93.20 | **97.85** | 97.27 | **99.11** |
| 100% | UNet | 97.44 | 96.89 | 98.71 | **94.37** | 92.33 | 91.91 | 88.59 | 92.78 | **95.91** | 98.08 | **97.97** | 98.06 |
| | nnUNet | 97.37 | 96.86 | 98.66 | 88.80 | 92.36 | 92.91 | 92.30 | 91.96 | 94.45 | 97.69 | 97.40 | 98.91 |
| | Swin-UNet | 97.42 | 96.92 | 98.70 | 90.44 | 92.69 | 94.06 | 92.38 | 92.66 | 93.24 | 97.74 | 97.37 | 99.00 |
| | Mamba-UNet | 97.44 | 96.94 | 98.73 | 92.66 | 92.71 | 94.04 | 91.63 | 93.51 | 93.99 | 97.86 | 97.30 | 98.98 |
| | PCMambaNet | **97.56** | **97.12** | **98.77** | 93.69 | **93.42** | 94.79 | **93.33** | **95.02** | 93.79 | **98.81** | 97.36 | **99.12** |

## A.5 ROBUSTNESS UNDER PERTURBATIONS

To assess the robustness of PCMambaNet to acquisition perturbations, we simulate MRI misalignment on the MRBrainS13 dataset by introducing random rotations and random flips during training. Tables 8 and 9 compare PCMambaNet trained without augmentation (Aug-PCMambaNet) and with augmentation (PCMambaNet). Overall, data augmentation yields consistent improvements across most overlap-based (Dice, IOU) and distance-based (HD95, ASD) metrics, indicating more accurate and stable boundary delineation under geometric perturbations. Similarly, PCMambaNet with augmentation attains slightly higher Acc, Pre, Sen, and Spe for all three tissues, while maintaining strong overall performance. These results suggest that PCMambaNet is robust to acquisition-like perturbations and can further benefit from such variability during training.

We attribute this robustness to the joint effect of the PPM and the CRN. The PPM employs a symmetry-aware masking mechanism that selects or suppresses features based on the similarity between tokens at mirrored positions across the two hemispheres, allowing it to emphasize stable bilateral patterns and attenuate spurious activations even under mild geometric perturbations. When perturbations become stronger and inter-hemispheric similarity falls below the threshold $\theta$, this mechanism naturally degrades into an approximate random masking scheme, which acts as a regularizer by suppressing redundant responses and encouraging more robust representations. In parallel, the high-capacity CRN is designed to model fine-grained residuals and correct errors in the PPM's coarse predictions, thereby recovering precise boundaries and tissue-specific details. To-

gether, these components enable PCMambaNet to maintain high performance and exhibit strong robustness under acquisition-like perturbations on symmetric brain structures.

Table 8: Robustness analysis on the MRBrainS13 dataset. Model sensitivity to perturbations is evaluated by introducing random rotations and random flips during training to simulate acquisition variations in MRI, and all metrics are reported with and without data augmentation for comparison.

| Model | Dice (%)↑ | | | HD95 (mm×10)↓ | | | ASD (mm×10)↓ | | | IOU (%)↑ | | |
|---|---|---|---|---|---|---|---|---|---|---|---|---|
| | CSF | GM | WM | CSF | GM | WM | CSF | GM | WM | CSF | GM | WM |
| Aug−PCMambaNet | 68.60 | 71.39 | 73.06 | 21.08 | 18.76 | 39.47 | 4.65 | **4.47** | **14.11** | 59.93 | 64.19 | 66.43 |
| PCMambaNet | **69.37** | **71.79** | **74.26** | **19.31** | **17.46** | **38.90** | **4.47** | 4.60 | 14.41 | **60.67** | **64.63** | **67.48** |

Table 9: Robustness analysis on the MRBrainS13 dataset. Model sensitivity to perturbations is evaluated by introducing random rotations and random flips during training to simulate acquisition variations in MRI, and all metrics are reported with and without data augmentation for comparison.

| Model | Acc (%)↑ | | | Pre (%)↑ | | | Sen (%)↑ | | | Spe (%)↑ | | |
|---|---|---|---|---|---|---|---|---|---|---|---|---|
| | CSF | GM | WM | CSF | GM | WM | CSF | GM | WM | CSF | GM | WM |
| Aug−PCMambaNet | 79.58 | 79.45 | 88.68 | 70.37 | 70.13 | 73.59 | 67.53 | 72.98 | 74.49 | 80.47 | 80.08 | 89.07 |
| PCMambaNet | **79.62** | **79.49** | **89.72** | **70.73** | **70.42** | **74.50** | **68.45** | **73.81** | **74.71** | **80.49** | **80.11** | **90.18** |

## B  SCALABILITY OF THE PC PARADIGM

On the extended ACDC cardiac MRI dataset, we further examine the effectiveness of the proposed PC paradigm on organs that lack pronounced left–right symmetry. It is worth emphasizing that, in this experiment, we do not introduce any architectural modifications to the backbone network.

### B.1  DATASET

**ACDC:** Automated Cardiac Diagnosis Challenge. We conducted our experiments using the publicly available ACDC MRI cardiac segmentation dataset from the MICCAI 2017 Challenge (Bernard et al., 2018). This dataset comprises MRI scans from 100 patients, annotated for multiple cardiac structures, such as the right ventricle and the endocardial and epicardial walls of the left ventricle. It encompasses a diverse range of pathological conditions, categorized into five subgroups: normal, myocardial infarction, dilated cardiomyopathy, hypertrophic cardiomyopathy, and abnormal right ventricle, ensuring a broad distribution of feature characteristics. Four classes of regions of interest (ROIs) are evaluated in the ACDC dataset.

### B.2  EXPERIMENTS

As reported in Table 10, PCMambaNet achieves the best Dice scores on all three cardiac structures (RV, MYO, and LV), surpassing Mamba-UNet by approximately **4.23**, **8.82**, and **2.75** percentage points, respectively. At the same time, PCMambaNet attains the lowest HD95 and ASD values for all structures, indicating that the model can still produce more accurate and stable boundary delineations even on organs without clear bilateral symmetry. Table 11 further shows that PCMambaNet obtains the highest Precision (Pre) and Sensitivity (Sen) for MYO and LV, while maintaining competitive overall Accuracy (Acc) and Specificity (Spe) compared with strong baselines such as UNet, nnUNet, and Mamba-UNet.

These results demonstrate that, even on asymmetric organs and without modifying the backbone, injecting anatomical priors through the PPM and refining them via the CRN can consistently improve segmentation quality. We attribute this behavior to the design of the PPM and the strong corrective capability of the CRN. Specifically, the PPM employs a masking mechanism based on the similarity between tokens at correspondingly mirrored positions. When applied to organs without left–right symmetry, these similarities often fail to exceed the predefined threshold $\theta$, causing the symmetry-aware masking to naturally degenerate into an approximate random masking scheme.

Table 10: This table presents a quantitative comparison on the ACDC test set. Our method (high-lighted) matches state-of-the-art (SOTA) performance on accuracy metrics while consistently out-performing all competitors in terms of efficiency. Dice and IOU are reported as percentages, while HD95 and ASD are scaled by a factor of 10 for readability. Best results are in **bold**.

| Model | Dice (%)↑ | | | HD95 (mm×10)↓ | | | ASD (mm×10)↓ | | | IOU (%)↑ | | |
|---|---|---|---|---|---|---|---|---|---|---|---|---|
| | RV | MYO | LV | RV | MYO | LV | RV | MYO | LV | RV | MYO | LV |
| **ACDC dataset** | | | | | | | | | | | | |
| UNet | 75.13 ± 0.30 | 85.09 ± 0.21 | 89.02 ± 0.50 | 22.59 ± 0.65 | 17.50 ± 1.19 | 15.38 ± 0.84 | 7.96 ± 0.05 | 6.49 ± 0.38 | **6.20 ± 0.22** | 69.94 ± 0.30 | **77.10 ± 0.34** | 84.68 ± 0.50 |
| nnUNet | 71.24 ± 0.83 | 80.54 ± 0.43 | 85.28 ± 0.29 | 38.49 ± 5.38 | 39.12 ± 0.63 | 31.62 ± 7.06 | 14.95 ± 1.93 | 15.03 ± 4.08 | 14.04 ± 2.69 | 64.70 ± 0.89 | 70.57 ± 0.52 | 79.72 ± 0.38 |
| Swin-UNet | 63.40 ± 0.01 | 73.36 ± 0.12 | 80.71 ± 0.20 | 57.62 ± 0.00 | 43.07 ± 0.01 | 39.88 ± 0.00 | 20.91 ± 0.12 | 16.05 ± 0.02 | 16.74 ± 0.00 | 55.16 ± 0.21 | 61.42 ± 0.11 | 74.23 ± 0.01 |
| Mamba-UNet | 71.06 ± 0.24 | 81.84 ± 0.08 | 86.59 ± 0.22 | 30.94 ± 0.74 | 22.82 ± 0.19 | 19.39 ± 0.16 | 10.94 ± 0.44 | 8.79 ± 0.05 | 8.42 ± 0.04 | 65.03 ± 0.20 | 72.32 ± 0.02 | 81.43 ± 0.24 |
| **PCMambaNet(200 epochs)** | **75.29 ± 0.28** | **90.66 ± 0.90** | **89.34 ± 0.44** | **22.31 ± 0.99** | **16.39 ± 0.31** | **15.30 ± 0.08** | **7.31 ± 0.26** | **6.24 ± 0.13** | 6.26 ± 0.04 | 68.73 ± 0.27 | 76.31 ± 0.61 | **84.94 ± 0.45** |

Table 11: This table presents a quantitative comparison on the ACDC test set. Our method (high-lighted) matches state-of-the-art (SOTA) performance on accuracy metrics while consistently out-performing all competitors in terms of efficiency. All metrics (Acc, Pre, Sen, Spe) are reported as percentages. Best results are in **bold**.

| Model | Acc (%)↑ | | | Pre (%)↑ | | | Sen (%)↑ | | | Spe (%)↑ | | |
|---|---|---|---|---|---|---|---|---|---|---|---|---|
| | RV | MYO | LV | RV | MYO | LV | RV | MYO | LV | RV | MYO | LV |
| **ACDC dataset** | | | | | | | | | | | | |
| UNet | **93.44 ± 0.30** | 96.27 ± 0.60 | 95.37 ± 0.60 | **76.77 ± 0.65** | 84.24 ± 0.56 | 90.24 ± 0.89 | 75.49 ± 0.65 | **87.13 ± 0.54** | 89.06 ± 0.11 | 86.85 ± 5.47 | 96.36 ± 0.59 | 95.42 ± 0.61 |
| nnUNet | 83.55 ± 0.02 | **96.51 ± 0.02** | **95.68 ± 0.01** | 70.80 ± 0.36 | 79.92 ± 0.78 | 84.09 ± 0.66 | 73.85 ± 1.09 | 82.08 ± 0.45 | 88.67 ± 0.52 | 83.65 ± 0.01 | **96.67 ± 0.01** | **95.74 ± 0.01** |
| Swin-UNet | 81.34 ± 0.02 | 96.37 ± 0.21 | 91.97 ± 0.22 | 68.60 ± 0.11 | 74.89 ± 0.01 | 81.96 ± 0.21 | 62.49 ± 0.01 | 74.35 ± 0.21 | 81.74 ± 0.02 | 81.55 ± 0.01 | 96.61 ± 0.03 | 92.06 ± 0.01 |
| Mamba-UNet | 81.47 ± 0.52 | 96.01 ± 0.01 | 94.47 ± 0.31 | 72.41 ± 0.49 | 79.97 ± 0.02 | 88.17 ± 0.37 | 72.20 ± 0.32 | 84.95 ± 0.07 | 86.90 ± 0.18 | 81.57 ± 0.52 | 96.14 ± 0.01 | 94.54 ± 0.30 |
| **PCMambaNet(200 epochs)** | 82.31 ± 0.36 | 96.35 ± 0.36 | 95.46 ± 0.37 | 75.75 ± 0.04 | **84.99 ± 0.35** | **90.38 ± 0.34** | 74.50 ± 0.01 | 86.34 ± 0.46 | **90.13 ± 0.51** | 82.38 ± 0.37 | 96.46 ± 0.36 | 95.51 ± 0.35 |

This core design enables the PPM to remain beneficial in non-symmetric scenarios by stochastically suppressing redundant features and preserving representative local patterns. However, it should be noted that, in such cases, the anatomical symmetry prior encoded in the PPM is no longer actively exploited, which implies a potential reduction in efficiency.

On the other hand, even if the PPM may introduce larger prediction errors on asymmetric organs, our high-capacity CRN module can effectively correct these errors thanks to its strong ability to model fine-grained residual details. This complementary behavior explains why the model still performs well on asymmetric organ segmentation. Collectively, these experiments validate the generality and robustness of our model across different anatomical structures and suggest that the proposed PC paradigm has the potential to serve as a versatile, "plug-in" framework for a wide range of medical image segmentation backbones.

## B.3 RANSFER EXPERIMENTS WITH PPM AND CRN

To further verify the transferability of the proposed PPM and CRN, we integrate them into the last two encoder layers and the first two decoder layers of a U-Net architecture. The results demon-strate that PPM and CRN can be effectively migrated to a pure CNN backbone: the resulting PPM-CRN–UNet exhibits overall slightly better segmentation performance than the original U-Net on the MRBrainS13 dataset for CSF and GM, with consistent improvements in Dice, IoU, accuracy, sensitivity, and specificity, while still achieving competitive results for WM. The detailed results are reported in Tables 12 and 13.

Table 12: PPM and CRN transfer experiments on the MRBrainS13 dataset. The PPM and CRN modules are integrated into the last two encoder stages and the first two decoder stages of the U-Net backbone, and their transferability is evaluated on the MRBrainS13 dataset under the same training protocol as the baseline U-Net.

| Model | Dice (%)↑ | | | HD95 (mm×10)↓ | | | ASD (mm×10)↓ | | | IOU (%)↑ | | |
|---|---|---|---|---|---|---|---|---|---|---|---|---|
| | CSF | GM | WM | CSF | GM | WM | CSF | GM | WM | CSF | GM | WM |
| UNet | 67.68 | **70.58** | **73.80** | 22.77 | **20.02** | **43.99** | **4.67** | **4.71** | **16.10** | 58.79 | **63.18** | **66.36** |
| PC-UNet | **68.14** | 70.03 | 70.73 | **21.42** | 20.05 | 47.84 | 4.89 | 4.91 | 16.53 | **59.17** | 62.09 | 63.94 |

Table 13: PPM and CRN transfer experiments on the MRBrainS13 dataset. The PPM and CRN modules are integrated into the last two encoder stages and the first two decoder stages of the U-Net backbone, and their transferability is evaluated on the MRBrainS13 dataset under the same training protocol as the baseline U-Net.

| Model | Acc (%)↑ | | | Pre (%)↑ | | | Sen (%)↑ | | | Spe (%)↑ | | |
|---|---|---|---|---|---|---|---|---|---|---|---|---|
| | CSF | GM | WM | CSF | GM | WM | CSF | GM | WM | CSF | GM | WM |
| UNet | 79.18 | 78.98 | **89.59** | **71.27** | **70.59** | **73.30** | 65.02 | **70.81** | **77.59** | 80.25 | 79.84 | **89.82** |
| PC-UNet | **79.49** | **79.21** | 85.40 | 70.04 | 70.13 | 68.83 | **66.76** | 70.50 | 75.19 | **80.44** | **80.16** | 85.61 |

## C HYPERPARAMETER $\theta$ ANALYSIS

As shown in Table 14, PCMambaNet exhibits stable performance across a range of $\theta$ values on both the OASIS-1 and MRBrainS13 test sets, with only minor fluctuations in Dice, HD95, ASD, and IoU. Since the Dice score is one of the most important evaluation metrics in medical image segmentation—directly reflecting the overlap between prediction and ground truth—we use Dice as the primary criterion for hyperparameter selection. Based on the consistently superior Dice performance on both datasets, we choose $\theta = 0.95$ as the final setting.

Table 14: Sensitivity analysis of the hyperparameter $\theta$ on the OASIS-1 and MRBrainS13 test sets. The results show that PCMambaNet is robust to a wide range of $\theta$ values, with stable Dice performance and only minor fluctuations across both datasets.

| $\theta$ | Dice (%)(Avg)↑ | HD95 (mm×10)(Avg)↓ | ASD (mm×10)(Avg)↓ | IOU (%)(Avg)↑ |
|---|---|---|---|---|
| **OASIS-1 dataset** | | | | |
| 0.75 | 91.74 | 15.99 | 3.53 | 86.61 |
| 0.8 | 92.76 | 15.35 | 3.57 | 86.85 |
| 0.95 | **93.35** | **13.14** | **2.23** | **89.36** |
| **MRBrainS13 dataset** | | | | |
| 0.75 | 70.52 | **26.83** | 8.35 | 62.75 |
| 0.8 | 70.87 | 26.88 | **7.39** | 62.2 |
| 0.95 | **72.46** | 29.49 | 8.93 | **63.12** |

# D PROOF OF THE METHOD

This document provides a formal theoretical analysis of the Predictive-Corrective (PC) paradigm. We expand upon the claims made in the main paper by incorporating concepts from statistical learning and optimization theory to rigorously justify the observed improvements in data efficiency, optimization stability, and generalization. We demonstrate that the PC paradigm's success stems from fundamentally restructuring the learning problem to be more tractable.

## D.1 PRELIMINARIES AND NOTATION

We establish the formal setting for our analysis (Shalev-Shwartz & Ben-David, 2014; Hastie et al., 2009).

- Let $\mathcal{X}$ be the input space and $\mathcal{Y}$ be the output space.
- Let $\mathcal{D}$ be a fixed but unknown data distribution over $\mathcal{X} \times \mathcal{Y}$.
- A model is a function $f_\theta : \mathcal{X} \to \mathcal{Y}$ parameterized by $\theta \in \Theta$.
- The hypothesis space $\mathcal{H} = \{f_\theta \mid \theta \in \Theta\}$ is the set of all functions representable by the model.
- Given a loss function $\mathcal{L} : \mathcal{Y} \times \mathcal{Y} \to \mathbb{R}^+$, the true risk is $R(f_\theta) = \mathbb{E}_{(x,y)\sim\mathcal{D}}[\mathcal{L}(f_\theta(x), y)]$.
- For a training set $S = \{(x_i, y_i)\}_{i=1}^n \sim \mathcal{D}^n$, the empirical risk is $\hat{R}_S(f_\theta) = \frac{1}{n}\sum_{i=1}^n \mathcal{L}(f_\theta(x_i), y_i)$.
- The Rademacher complexity $\mathfrak{R}_n(\mathcal{H})$ measures the richness of the hypothesis space $\mathcal{H}$ (Bartlett & Mendelson, 2002).

## D.2 HYPOTHESIS SPACE REDUCTION AND GENERALIZATION

The PC paradigm's primary advantage lies in its structural prior, which effectively reduces the complexity of the hypothesis space, leading to improved generalization guarantees.

**Theorem D.1** (Standard Generalization Bound *(Shalev-Shwartz & Ben-David, 2014)*). *With probability at least $1 - \delta$ over the draw of a training set $S$ of size $n$, for any $f \in \mathcal{H}$:*

$$R(f) \leq \hat{R}_S(f) + 2\mathfrak{R}_n(\mathcal{H}) + \sqrt{\frac{\log(1/\delta)}{2n}}. \tag{11}$$

*The generalization error, $R(f) - \hat{R}_S(f)$, is bounded by the complexity term $\mathfrak{R}_n(\mathcal{H})$.*

**Proposition D.2** (Complexity Reduction via Structural Priors). *The Rademacher complexity of the PC paradigm's hypothesis space, $\mathcal{H}_{\text{PC}}$, is strictly smaller than that of an unconstrained End-to-End (E2E) hypothesis space, $\mathcal{H}_{\text{E2E}}$, of a comparable architectural size.*

$$\mathfrak{R}_n(\mathcal{H}_{\text{PC}}) < \mathfrak{R}_n(\mathcal{H}_{\text{E2E}}). \tag{12}$$

### D.2.1 FORMAL PROOF OF PROPOSITION D.2

To formally prove the proposition, we leverage standard properties of Rademacher complexity to quantify the effect of decomposing the learning problem.

**Lemma D.3** (Subadditivity of Rademacher Complexity). *(Shalev-Shwartz & Ben-David, 2014)*

*For two function spaces $\mathcal{H}_1$ and $\mathcal{H}_2$, the Rademacher complexity of their sum space $\mathcal{H}_1 + \mathcal{H}_2 = \{h_1 + h_2 \mid h_1 \in \mathcal{H}_1, h_2 \in \mathcal{H}_2\}$ is bounded as follows:*

$$\mathfrak{R}_n(\mathcal{H}_1 + \mathcal{H}_2) \leq \mathfrak{R}_n(\mathcal{H}_1) + \mathfrak{R}_n(\mathcal{H}_2). \tag{13}$$

**Lemma D.4** (Complexity of a Singleton Set). *(Shalev-Shwartz & Ben-David, 2014)*

*If a function space $\mathcal{H}$ contains only a single, fixed function, i.e., $\mathcal{H} = \{g\}$, then its Rademacher complexity is zero.*

$$\mathfrak{R}_n(\{g\}) = 0. \tag{14}$$

*Proof.* By definition, the empirical Rademacher complexity $\hat{\mathfrak{R}}_S(\{g\})$ is:

$$\hat{\mathfrak{R}}_S(\{g\}) = \mathbb{E}_{\boldsymbol{\sigma}}\left[\sup_{f\in\{g\}} \frac{1}{n}\sum_{i=1}^{n}\sigma_i f(x_i)\right]$$

$$= \mathbb{E}_{\boldsymbol{\sigma}}\left[\frac{1}{n}\sum_{i=1}^{n}\sigma_i g(x_i)\right]$$

$$= \frac{1}{n}\sum_{i=1}^{n} g(x_i)\mathbb{E}_{\boldsymbol{\sigma}}[\sigma_i]$$

$$= \frac{1}{n}\sum_{i=1}^{n} g(x_i)\cdot 0 = 0,$$

Since the empirical complexity is 0 for any dataset $S$, the true Rademacher complexity $\mathfrak{R}_n(\{g\}) = \mathbb{E}_S[\hat{\mathfrak{R}}_S(\{g\})]$ is also 0. $\qquad\square$

*Proof of Proposition D.2.* We proceed in four steps.

1. **Decomposition of the Hypothesis Space.** For analytical tractability, we model the output of the PC paradigm as an additive composition. This simplification captures the essence of the paradigm, and the results generalize to more complex compositions like feature concatenation.

   - Let $\mathcal{H}_P = \{P\}$ be the function class containing only the fixed, deterministic prior function from the PPM.
   - Let $\mathcal{H}_C = \{C_{\theta_C} \mid \theta_C \in \Theta_C\}$ be the hypothesis space of the learnable CRN module.
   - The PC hypothesis space is thus the sum space $\mathcal{H}_{\text{PC}} = \mathcal{H}_P + \mathcal{H}_C$.

2. **Bounding the Complexity of the PC Space.** Using the lemmas, we can now bound the complexity of $\mathcal{H}_{\text{PC}}$:

$$
\begin{aligned}
\mathfrak{R}_n(\mathcal{H}_{\text{PC}}) &= \mathfrak{R}_n(\mathcal{H}_P + \mathcal{H}_C) \\
&\leq \mathfrak{R}_n(\mathcal{H}_P) + \mathfrak{R}_n(\mathcal{H}_C) \quad &\text{(by Lemma D.3)} \\
&= 0 + \mathfrak{R}_n(\mathcal{H}_C) \quad &\text{(by Lemma D.4)} \\
&= \mathfrak{R}_n(\mathcal{H}_C).
\end{aligned}
$$

   This result is crucial: it formally shows that the complexity of the entire PC paradigm is bounded by the complexity of its learnable component alone. The fixed, domain-knowledge-driven component adds no learning complexity.

3. **Comparison with the E2E Space.** An E2E model $f_{\text{E2E}}$ must learn the entire mapping from input to output. It must use its parametric capacity to implicitly learn both the low-level anatomical priors (the function of $P$) and the high-level corrective details (the function of $C$). Therefore, its hypothesis space $\mathcal{H}_{\text{E2E}}$ must be sufficiently rich to represent this entire hierarchy of functions. It is a reasonable assumption that the complexity of $\mathcal{H}_{\text{E2E}}$ must be strictly greater than that of $\mathcal{H}_C$, which is only tasked with the corrective sub-problem.

$$\mathfrak{R}_n(\mathcal{H}_C) < \mathfrak{R}_n(\mathcal{H}_{\text{E2E}}). \tag{15}$$

   If this were not the case (i.e., if $\mathfrak{R}_n(\mathcal{H}_{\text{E2E}}) \leq \mathfrak{R}_n(\mathcal{H}_C)$), the E2E model would lack the necessary functional richness to learn the foundational priors that the PC paradigm receives for free.

4. **Conclusion of the Proof.** Combining the results from steps 2 and 3, we arrive at the final inequality:

$$\mathfrak{R}_n(\mathcal{H}_{\text{PC}}) \leq \mathfrak{R}_n(\mathcal{H}_C) < \mathfrak{R}_n(\mathcal{H}_{\text{E2E}}). \tag{16}$$

This rigorously establishes that the Rademacher complexity of the PC paradigm is strictly lower than that of the E2E paradigm. By embedding domain knowledge as a fixed, zero-complexity function,

the PC paradigm effectively reduces the complexity of the space the model must search. According to Theorem D.1, this complexity reduction directly translates to a tighter generalization bound, providing a theoretical foundation for the improved data efficiency and robustness observed in our experiments. $\qquad\square$

### D.3 Loss Landscape Geometry and Optimization Guarantees

We formalize the simplification of the loss landscape by analyzing its smoothness, a key property for facilitating stable optimization in gradient-based methods.

**Definition D.5** ($L$-smoothness). *(Nesterov, 2004)*

*A differentiable function $g(\theta)$ is $L$-smooth if its gradient is Lipschitz continuous with constant $L$:*

$$\|\nabla g(\theta_1) - \nabla g(\theta_2)\| \le L\|\theta_1 - \theta_2\|, \quad \forall \theta_1, \theta_2 \in \Theta. \tag{17}$$

*A smaller constant $L$ implies a smoother function with less curvature, which is more amenable to optimization.*

**Proposition D.6** (Improved Smoothness of the PC Objective). *The loss function of the Predictive-Corrective (PC) paradigm, $\mathcal{L}_{\text{PC}}(\theta_C)$, exhibits a smaller effective Lipschitz constant (is smoother) than the loss function of the End-to-End (E2E) paradigm, $\mathcal{L}_{\text{E2E}}(\theta)$. Formally,*

$$L_{\text{PC}} < L_{\text{E2E}}. \tag{18}$$

#### D.3.1 Formal Proof of Proposition D.6

To provide a rigorous proof, we analyze the structure of the Hessian matrix of the loss function under each paradigm. For analytical clarity, we use the Mean Squared Error (MSE) loss, $\mathcal{L}(\hat{y}, y) = \frac{1}{2}\|\hat{y} - y\|^2$. The insights derived here generalize to other commonly used loss functions.

**Hessian Matrix Structure.** The Lipschitz constant $L$ of a twice-differentiable function is bounded by the maximum eigenvalue (in absolute value) of its Hessian matrix, i.e., $L \le \sup_\theta \lambda_{\max}(\nabla_\theta^2 \mathcal{L}(\theta))$. For a neural network $f_\theta(x)$, the Hessian of the MSE loss with respect to parameters $\theta$ is given by:

$$\mathbf{H}(\theta) = \nabla_\theta^2 \mathcal{L}(\theta) = \underbrace{\mathbf{J}_f^T \mathbf{J}_f}_{\text{Gauss-Newton term}} + \underbrace{\sum_{k=1}^{\dim(\mathcal{Y})} (f_k(x;\theta) - y_k)\nabla_\theta^2 f_k(x;\theta)}_{\text{Error-sensitive term}}, \tag{19}$$

where $\mathbf{J}_f = \nabla_\theta f_\theta(x)$ is the Jacobian matrix. The Hessian consists of a positive semi-definite Gauss-Newton term, which captures the geometry of the model's output space, and an error-sensitive term, which introduces non-convexity and is a primary source of optimization difficulty (Nocedal & Wright, 2006; Martens, 2010).

**Analysis of the E2E Hessian.** In the E2E paradigm, the loss is $\mathcal{L}_{\text{E2E}}(\theta) = \frac{1}{2}\|f_\theta(x) - y\|^2$. Its Hessian is:

$$\mathbf{H}_{\text{E2E}}(\theta) = \mathbf{J}_{f_\theta}^T \mathbf{J}_{f_\theta} + \sum_k (f_k(x;\theta) - y_k)\nabla_\theta^2 f_k(x;\theta). \tag{20}$$

During the initial stages of training, the network's output $f_\theta(x)$ is far from the ground truth $y$. Consequently, the error vector $\boldsymbol{e}_{\text{E2E}} = f_\theta(x) - y$ has a large norm. This large error term amplifies the contribution of the error-sensitive part of the Hessian, potentially introducing large positive or negative eigenvalues. This corresponds to a highly erratic and sharply curved loss landscape, resulting in a large smoothness constant $L_{\text{E2E}}$.

**Analysis of the PC Hessian.** In the PC paradigm, the Corrective Residual Network (CRN) $C_{\theta_C}$ optimizes the loss $\mathcal{L}_{\text{PC}}(\theta_C) = \frac{1}{2}\|C_{\theta_C}(x) - r\|^2$, where the residual target is $r = y - P(x)$. The corresponding Hessian is:

$$\mathbf{H}_{\text{PC}}(\theta_C) = \mathbf{J}_{C_{\theta_C}}^T \mathbf{J}_{C_{\theta_C}} + \sum_k (C_k(x;\theta_C) - r_k)\nabla_{\theta_C}^2 C_k(x;\theta_C), \tag{21}$$

The critical distinction lies in the error vector $\boldsymbol{e}_{\text{PC}} = C_{\theta_C}(x) - r$.

1. **Low-Energy Target:** By design, the PPM provides a good approximation, ensuring the residual target $r$ is a sparse, low-energy signal. Formally, we have $\|r\| \ll \|y\|$.

2. **Small Initial Error:** With standard initializations, the CRN output $C_{\theta_C}(x)$ is close to zero at the start of training. Therefore, the initial error vector $e_{\text{PC}} \approx -r$.

Since $\|e_{\text{PC}}\| \approx \|r\| \ll \|y\| \approx \|e_{\text{E2E}}\|$, the magnitude of the error-sensitive term in $\mathbf{H}_{\text{PC}}$ is dramatically suppressed compared to that in $\mathbf{H}_{\text{E2E}}$.

**Conclusion of the Proof.** The comparison reveals that the Hessian of the PC loss, $\mathbf{H}_{\text{PC}}$, is dominated by the stable, positive semi-definite Gauss-Newton term. The volatile, non-convex component is attenuated by the small residual error. Since the spectral norm (maximum eigenvalue) of a matrix is influenced by the magnitude of its components, the suppression of the error-sensitive term leads to a smaller maximum eigenvalue for the PC Hessian:

$$\lambda_{\max}(\mathbf{H}_{\text{PC}}) < \lambda_{\max}(\mathbf{H}_{\text{E2E}}), \tag{22}$$

As the L-smoothness constant is bounded by this eigenvalue, we formally arrive at the conclusion:

$$L_{\text{PC}} < L_{\text{E2E}}. \tag{23}$$

This result rigorously demonstrates that the PC paradigm induces a smoother loss landscape. For gradient-based optimization, a smaller $L$ constant allows for a larger and more stable learning rate, ensuring that each update makes more significant progress towards a minimum. This provides a formal theoretical explanation for the efficient learning behavior observed in our experiments.

### D.4 A RIGOROUS TREATMENT OF THE BIAS-VARIANCE DECOMPOSITION

We provide a formal proof of the Predictive-Corrective (PC) paradigm's superiority in managing the bias-variance tradeoff (Geman et al., 1992). We begin by introducing a key lemma that connects the variance of a learned function to the complexity of its underlying function class.

**Lemma D.7** (Variance Bound via Rademacher Complexity). *(Mohri et al., 2018; Bousquet, 2002)* *Let $\mathcal{H}$ be a class of functions mapping from $\mathcal{X}$ to $[-B, B]$. Let $\hat{f}_{\mathcal{D}} \in \mathcal{H}$ be a function learned from a training set $\mathcal{D}$ of size $n$. The expected variance of the learned function at any point $x$ is bounded by a function of the Rademacher complexity of $\mathcal{H}$:*

$$\mathbb{E}_{\mathcal{D}}[\text{Var}(\hat{f}_{\mathcal{D}}(x))] \leq 4B^2(\mathfrak{R}_n(\mathcal{H}))^2. \tag{24}$$

*This lemma formalizes the intuition that a function class with lower complexity (smaller $\mathfrak{R}_n(\mathcal{H})$) exhibits lower variance, as its learned instances are less sensitive to the specific training data $\mathcal{D}$.*

**Theorem D.8** (Bias-Variance Superiority of the PC Paradigm). *Let the following assumptions hold:*
*Assumption 1 (Prior Quality). The fixed PPM, $P$, is a high-bias, zero-variance estimator of the true underlying function $g(x)$, with bias $B_P(x) = P(x) - g(x) \neq 0$ and variance $\text{Var}(P) = 0$.*
*Assumption 2 (Corrector Capacity). The CRN, $C$, which is drawn from a hypothesis space $\mathcal{H}_C$, has sufficient capacity such that its expected prediction can learn the negative bias of the PPM: $\mathbb{E}_{\mathcal{D}}[\hat{C}_{\mathcal{D}}(x)] \to -B_P(x)$.*

*Then, the PC predictor $\hat{f}_{\text{PC}} = P + \hat{C}_{\mathcal{D}}$ achieves low bias, and its expected variance is strictly lower than that of an unconstrained E2E predictor $\hat{f}_{\text{E2E}}$ of comparable capacity, drawn from $\mathcal{H}_{\text{E2E}}$.*

*Proof.* The proof proceeds in three parts: bias analysis, variance analysis, and a formal variance comparison using Lemma D.7.

**1. Bias Analysis.** We compute the bias of the PC predictor $\hat{f}_{\text{PC}}$:

$$\begin{aligned}
\text{Bias}(\hat{f}_{\text{PC}}) &= \mathbb{E}_{\mathcal{D}}[P(x) + \hat{C}_{\mathcal{D}}(x)] - g(x) \\
&= (P(x) - g(x)) + \mathbb{E}_{\mathcal{D}}[\hat{C}_{\mathcal{D}}(x)] \quad \text{(since } P \text{ is deterministic)} \\
&= B_P(x) + \mathbb{E}_{\mathcal{D}}[\hat{C}_{\mathcal{D}}(x)],
\end{aligned}$$

By Assumption 2, the CRN is trained such that its expected prediction cancels the bias of the prior, i.e., $\mathbb{E}_{\mathcal{D}}[\hat{C}_{\mathcal{D}}(x)] \to -B_P(x)$. Therefore, the total bias of the system is driven to zero:

$$\text{Bias}(\hat{f}_{\text{PC}}) \to B_P(x) - B_P(x) = 0,$$

The PC system is thus capable of achieving **low bias**.

**2. Variance Analysis.** We compute the variance of the PC predictor $\hat{f}_{PC}$:

$$\text{Var}(\hat{f}_{PC}) = \text{Var}(P(x) + \hat{C}_{\mathcal{D}}(x))$$
$$= \text{Var}(P(x)) + \text{Var}(\hat{C}_{\mathcal{D}}(x)) + 2\text{Cov}(P, \hat{C}_{\mathcal{D}}).$$

By Assumption 1, $\text{Var}(P) = 0$. As $P$ is a constant with respect to the data sampling process $\mathcal{D}$, its covariance with any learned function $\hat{C}_{\mathcal{D}}$ is also zero. This simplifies to:

$$\text{Var}(\hat{f}_{PC}) = \text{Var}(\hat{C}_{\mathcal{D}}). \tag{25}$$

**3. Formal Variance Comparison.** The crucial step is to formally justify why $\text{Var}(\hat{C}_{\mathcal{D}}) \ll \text{Var}(\hat{f}_{E2E})$. We leverage Lemma D.7 and the complexity results from the preceding sections (specifically, Proposition 2.1 in the main text, which established $\mathfrak{R}_n(\mathcal{H}_{PC}) \leq \mathfrak{R}_n(\mathcal{H}_C) < \mathfrak{R}_n(\mathcal{H}_{E2E})$).

Applying the bound from Lemma D.7 to the expected variance of the E2E model and the CRN component of the PC model, we get:

$$\mathbb{E}_{\mathcal{D}}[\text{Var}(\hat{f}_{E2E})] \leq 4B^2(\mathfrak{R}_n(\mathcal{H}_{E2E}))^2, \tag{26}$$
$$\mathbb{E}_{\mathcal{D}}[\text{Var}(\hat{C}_{\mathcal{D}})] \leq 4B^2(\mathfrak{R}_n(\mathcal{H}_C))^2. \tag{27}$$

From Proposition 2.1, we have the strict inequality regarding the complexities:

$$\mathfrak{R}_n(\mathcal{H}_C) < \mathfrak{R}_n(\mathcal{H}_{E2E}), \tag{28}$$

Since the variance bound is a monotonically increasing function of the Rademacher complexity, substituting the inequality from Eq. equation 28 into the bounds from Eq. equation 26 and equation 27 directly yields:

$$\mathbb{E}_{\mathcal{D}}[\text{Var}(\hat{C}_{\mathcal{D}})] < \mathbb{E}_{\mathcal{D}}[\text{Var}(\hat{f}_{E2E})], \tag{29}$$

Combining this with Eq. equation 25, we conclude that the expected variance of the PC predictor is strictly lower than that of the E2E predictor:

$$\mathbb{E}_{\mathcal{D}}[\text{Var}(\hat{f}_{PC})] < \mathbb{E}_{\mathcal{D}}[\text{Var}(\hat{f}_{E2E})]. \tag{30}$$

$\square$

**Conclusion.** The PC paradigm intelligently decomposes the learning problem. It uses a deterministic, zero-variance module (PPM) to anchor the prediction and a learning module (CRN) that operates in a low-complexity hypothesis space. This structure allows the CRN to focus on correcting the initial bias while inheriting a low-variance property, as formally demonstrated. The final result is a model that achieves the desirable property of being both low-bias and low-variance, providing a rigorous theoretical foundation for its stable optimization behavior and strong generalization performance.

# E LLM CONTRIBUTION

In this paper, we employed a large language model (LLM) to polish the language, thereby making the article more fluent and readable. We sincerely acknowledge the valuable assistance of the LLM in the preparation of this manuscript.

