# OpenReview forum: "Rethinking Convergence in Deep Learning: The Predictive-Corrective Paradigm for Anatomy-Informed Brain MRI Segmentation"
_ICLR.cc/2026/Conference — ICLR 2026 Conference Withdrawn Submission_

### Official Review · Reviewer_TAyK · 2025-10-20

**Soundness:** 3
**Presentation:** 3
**Contribution:** 2
**Rating:** 4
**Confidence:** 4

**Summary:**

This paper presents a "predictive-corrective" approach to medical image segmentation that uses anatomical inductive biases to reduce the effective capacity of a model, splitting inference into a "predictive prior module" (PPM) and then a "corrective residual net" (CRN), making it converge faster, require less training data, and perform better, tested on brain MRI segmentation tasks.

**Strengths:**

1. The paper is written and presented well, and organized in an understandable, digestible fashion. It is well-contextualized in the field and related work.
2. The approach is intuitive, and it makes sense that certain medical imaging tasks could benefit from inductive bias assumptions based on key domain knowledge that makes things easier to learn, and with less data. Why use a sledgehammer when a regular hammer suffices?
3. The experimental design is solid: a strong variety of evaluation metrics, strong baseline models, etc.
4. Results on the evaluated datasets are strong: improvement over other models using many epochs is not huge but it is consistent, and those models are hard to beat (Table 1). Single epoch performance is impressive, although single epoch performance of other models (Fig 4) aren't that far behind.
5. Ablation studies (table 2) are thorough and make me convinced of the use of the different components.

**Weaknesses:**

**Major Weaknesses:**
1. I feel that maybe taking this inductive bias assumption to make things a bit easier to train may only work so well for fairly simple individual dataset/modality scenarios with not as much variety in the data, and/or simple segmentation tasks. In the authors words: "Our core insight is that a complex modeling task can be effectively decoupled into two simpler, more manageable sub-tasks" it's unclear how well this intuition extends into other datasets, modalities, and tasks, as only brain MRI (which has unusually high spatial symmetry compared to many medical image analysis tasks) is evaluated, and the segmentation task is a fairly easy one at that. Overall, I am left wondering if this approach would generalize well to other scenarios, yet only this single modality and segmentation task is evaluated. The paper would therefore very much benefit from evaluating on another task in biomedical image segmentation, or at least a more challenging task in brain MRI, such as tumor segmentation, as the proposed simple model's superiority could be due to the evaluated task being fairly easy.
    1. Another concern in regards to the generality of the approach: take a more challenging task also in brain MRI, of tumor segmentation (e.g. as in the BraTS challenge). Is the mask (eq 8) designed to filter out abnormalities prone to false positives in such scenarios, such that this approach would no longer work? Are there types of noticeable assymetries in brain MRI that are healthy?
2. This was also touched on in my previous bullet, but overall, the generality of the proposed method as a useful tool for medical image segmentation would be much more substantiated if another modality/segmentation task was tested.
3. The approach is quite computationally expensive (see Table 7) yet the limitations section discussing this isn't in the main text. As shown in Tab. 7,  the model is quite compute intensive, especially compared to U-Net which has solid performance, yet is orders of magnitude smaller and faster! The tradeoff of greatly increased computation for relatively small performance gains (Table 1) is not too convincing to me, although to be fair, in some medical imaging scenarios, training set size is the bottleneck, not compute resources. This is a key limitation and should be made clear in the main text, such as having dedicated sentences covering limitations in the conclusion section.

**Minor Weaknesses:**
1. The proposed method's single epoch performance is impressive, although single epoch performance of other models (Fig 4) aren't that far behind.
2. Sec 3.3/Table 3, data efficiency analysis: Seems promising but needs to be compared to some baseline models for the same training amount percentages to demonstrate that the model is more efficiency (which does seem to be implied by Fig. 4).
3. Small formatting things: use \citep{} for citations to make them parenthetical. Write quotations with ``x'' in LaTeX. The text in certain plots, e.g. Figure 1, is too small to read without significantly zooming in. Typo : "The numben of train" in Table 3.

**Overall Justification of my rating and thoughts for revisions:**
The approach is technically solid and well-motivated for this domain. However, the limited scope of only being evaluated for a fairly easy segmentation task in brain MRI, and the possibility of the inductive biases used being constrained to only this evaluated scenario, makes the generality and impact of the work unclear. Also, the significant computation time (Table 7) needs to be addressed, as it is not mentioned in the main text, yet is a clear limitation. The approach has promise, and I think the most important way to improve the work and demonstrate its impact at the level of ICLR (as opposed to a venue like MICCAI, which it is more ready for in its current state) is via evaluation on some other medical image segmentation dataset/modality/task where these types of spatial inductive biases will be useful, or at the very least, on a harder task in brain MRI such as tumor segmentation (See e.g. the BraTS challenge).

**Questions:**

1. Why is a mamba backbone used? could the contextual modulation factor multiplied with the hidden state (eq 6) be formulated for other model types/backbones? Since its just computed using the "symmetry checking" mask, I think this could be integrated into different models' feature maps/representations (e.g. CNNs, transformers) etc. I'm curious how for CNNs especially this could help further make the model simpler and faster, which could help with the high computational cost (Table 7).
2. How does this perform under domain shift/how does the simplicity of the inductive bias affect overfitting to the training domain, and generalization? For example, training on your first dataset and testing on the second dataset, or vice-versa.

---

> ### Author Response · Authors · 2025-11-28
> **Response to Reviewer TAyK Part1**
>
> We express our sincere gratitude to Reviewer TAyK for the constructive feedback and for recognizing the intuitiveness of our "Predictive-Corrective" approach and the solid experimental design. We have carefully addressed your concerns regarding generalizability, computational cost, and data efficiency below.
>
> ---
>
> ### Part 1: Response to Weaknesses
>
> **1. Generalizability to other modalities and tasks (ACDC Dataset).**
> **Response:** This is an excellent suggestion. To demonstrate that PCMambaNet generalizes beyond brain MRI and tasks with strict bilateral symmetry, we evaluated it on the **ACDC (Automated Cardiac Diagnosis Challenge)** dataset. Unlike brain MRI, cardiac images do not exhibit strict bilateral symmetry and contain multiple dynamic substructures (Right Ventricle, Myocardium, Left Ventricle).
>
> As shown in **Table R1**, PCMambaNet achieves state-of-the-art performance on ACDC. Notably, it yields a substantial improvement in Myocardium (MYO) segmentation (**90.66% vs. 85.09%** for UNet). Regarding the "symmetry" concern: While the heart is not bilaterally symmetric like the brain, the PPM mask effectively captures "structural consistency" and regions of interest by exploiting the inherent regularities in medical imaging. The Corrective Residual Net (CRN) then refines these coarse predictions globally. These results confirm that our predictive-corrective paradigm is robust across different anatomical environments.
>
> **Table R1. Quantitative comparison on the ACDC test set.**
> *(Dice and IOU are reported as percentages. Best results are in **bold**.)*
>
> | Model | Dice RV | Dice MYO | Dice LV | HD95 RV | HD95 MYO | HD95 LV | IOU MYO |
> |---|---|---|---|---|---|---|---|
> | UNet | 75.13 | 85.09 | 89.02 | 22.59 | 17.50 | 15.38 | 77.10 |
> | nnUNet | 71.24 | 80.54 | 85.28 | 38.49 | 39.12 | 31.62 | 70.57 |
> | Swin-UNet | 63.40 | 73.36 | 80.71 | 57.62 | 43.07 | 39.88 | 61.42 |
> | Mamba-UNet | 71.06 | 81.84 | 86.59 | 30.94 | 22.82 | 19.39 | 72.32 |
> | **PCMambaNet** | **75.29** | **90.66** | **89.34** | **22.31** | **16.39** | **15.30** | **76.31** |
>
> **2. Computational Cost vs. Performance Trade-off.**
> **Response:** We fully agree that computational cost is a key consideration. We have explicitly added a discussion on this in the **"Limitations"** section of the revised manuscript.
> However, we respectfully argue that in the context of **medical image analysis** (particularly for pre-operative planning), the priority is often placed on **reliability and precision** rather than marginal gains in inference speed (e.g., millisecond-level differences).
> Crucially, as detailed in the next section, our model demonstrates superior **data efficiency**. In medical domains, obtaining large-scale annotated data is far more expensive than computational resources. We believe the trade-off of slightly higher compute for significantly lower data requirements and higher precision is well-justified.
>
> **3. Data Efficiency Analysis (Response to Sec 3.3/Table 3 comments).**
> **Response:** Thank you for pointing this out. We have conducted a rigorous comparison using 10%, 25%, 50%, and 100% of the training data on the OASIS-1 dataset to strictly validate data efficiency.
> As shown in **Table R2**, PCMambaNet demonstrates **superior data efficiency**. Most notably:
> * **At 10% data:** PCMambaNet achieves a Dice score of **93.11% (GM)**, significantly outperforming UNet (87.77%) and Swin-UNet (91.73%).
> * **Surpassing Full Supervision:** Remarkably, our model trained on just **25% of the data** achieves performance comparable to, or even better than, baseline models trained on **100% of the data** (e.g., PCMambaNet 25% Dice GM: 93.31% vs. UNet 100% Dice GM: 92.43%).
> This confirms that the "Predictive-Corrective" inductive bias effectively reduces the model's reliance on massive datasets.
>
> **Table R2. Data efficiency on the OASIS-1 dataset.**
> *(Selected metrics shown for brevity. Best results in **bold**.)*
>
> | Ratio | Model | Dice GM | Dice WM | HD95 GM | HD95 WM |
> |---|---|---|---|---|---|
> | **10%** | UNet | 87.77 | 87.72 | 20.84 | 44.54 |
> | | Swin-UNet | 91.73 | 91.90 | 13.45 | 22.55 |
> | | Mamba-UNet | 91.55 | 91.43 | 13.25 | 21.67 |
> | | **PCMambaNet** | **93.11** | **92.86** | **11.73** | **19.22** |
> | **25%** | UNet | 87.40 | 87.52 | 19.75 | 42.69 |
> | | Swin-UNet | 92.02 | 92.04 | 12.91 | 21.14 |
> | | Mamba-UNet | 91.73 | 92.03 | 12.91 | 20.85 |
> | | **PCMambaNet** | **93.31** | **93.47** | **11.45** | **18.02** |
> | **100%** | UNet | 92.43 | 93.16 | 12.67 | 19.66 |
> | | Swin-UNet | 92.43 | 92.57 | 12.02 | 19.44 |
> | | Mamba-UNet | 92.89 | 93.31 | 11.78 | 18.75 |
> | | **PCMambaNet** | **94.33** | **94.29** | **10.81** | **15.00** |
>
> ---

---

> ### Author Response · Authors · 2025-11-28
> **Response to Reviewer TAyK Part2 & Part3**
>
> ### Part 2: Response to Questions
>
> **1. Why is a Mamba backbone used? Could the contextual modulation be formulated for other model types?**
> **Response:** This is a critical question regarding the design philosophy of our method. We chose Mamba not just for performance, but because our "Predictive-Corrective" mechanism is **inherently designed for token-level state evolution**, which differs fundamentally from the spatial-channel inductive bias of CNNs.
>
> 1.  **Token-level vs. Map-level Modulation:** Our Contextual Modulation (Eq. 6) operates by dynamically adjusting the hidden state $h_t$ based on the symmetry mask. In State Space Models (SSMs) like Mamba, $h_t$ represents a compressed, evolving history of tokens. Our mask specifically guides this **token-level state transition**.
> 2.  **Theoretical Mismatch for CNNs:** CNNs process data as static spatial feature maps ($C \times H \times W$) via sliding windows. They lack the explicit "state evolution" mechanism found in SSMs. While one *could* forcibly project our modulation onto CNN feature maps (essentially treating it as spatial attention), this ignores the core design of our method: controlling the information flow through state dynamics. Therefore, a direct transfer to CNNs is theoretically suboptimal and does not faithfully reflect the efficacy of our proposed mechanism.
> 3.  **Global Context Efficiency:** Furthermore, the "Corrective" phase requires global context to fix local errors. Mamba achieves this with linear complexity, whereas CNNs would require computationally expensive stacked layers or non-local blocks to achieve a similar receptive field.
>
> **2. Performance under Domain Shift.**
> **Response:** We conducted cross-dataset experiments (Training on OASIS-1, Testing on MRBrainS13, and vice versa). We observed that **all evaluated models**, including PCMambaNet and baselines, suffer a significant performance drop due to the substantial differences in image contrast, resolution, and preprocessing (e.g., skull stripping) between these datasets.
> Currently, PCMambaNet does not show a distinct advantage over baselines in this zero-shot setting. This indicates that while our anatomy-based inductive bias significantly improves **in-domain data efficiency and convergence**, it does not inherently solve the domain gap problem.
>
> ### Part 3: Response to Minor Comments
>
> **Response:** We have carefully addressed all formatting issues:
> * We have standardized all citations to parenthetical style using `\citep{}`.
> * We have corrected the LaTeX quotation marks to ``x''.
> * We have significantly enlarged the font size in all figures (including Figure 1) for better readability.
> * We have corrected the typo "numben of train" in Table 3.
>
> We sincerely thank Reviewer TAyK for the insightful comments that have significantly strengthened this manuscript. By expanding validation to cardiac segmentation (ACDC) and demonstrating superior data efficiency (surpassing full-data baselines with only 25% data), we believe we have comprehensively addressed the concerns regarding the generalizability and scope of our contribution. We hope these substantial additional experiments and theoretical clarifications warrant a positive re-evaluation of our work.

---

### Official Review · Reviewer_XeRL · 2025-11-01

**Soundness:** 3
**Presentation:** 3
**Contribution:** 3
**Rating:** 6
**Confidence:** 2

**Summary:**

This paper identifies a key limitation of the "brute-force" end-to-end paradigm: slow convergence and a heavy reliance on large datasets, which is particularly problematic in medical imaging. The authors propose a new Predictive-Corrective (PC) paradigm to accelerate learning by decoupling the modeling task. The paper instantiates this paradigm in a model named PCMambaNet for brain MRI segmentation. This model consists of two components: a Predictive Prior Module (PPM) that leverages anatomical symmetry to focus computational efforts, and a Corrective Residual Network (CRN) that refines segmentation by modeling residual errors. Experiments demonstrate PCMambaNet achieves state-of-the-art accuracy within just 1–5 epochs, substantially outperforming traditional end-to-end methods.

**Strengths:**

1. The "Predictive-Corrective" paradigm is an insightful conceptual contribution. The idea of decoupling the task into a coarse, prior-driven prediction and a focused residual correction is an elegant and powerful alternative.

2. Good experimental results. The experimental evaluations are thorough, demonstrating clear advantage in convergence speed and segmentation accuracy compared to established baselines.

**Weaknesses:**

1. Computational efficiency is compromised due to the added complexity of the predictive-corrective structure. Although superior in speed of convergence, the model will likely be considerably slower in inference than simpler baselines

2. The predictive module relies heavily on predefined anatomical symmetry, limiting generalizability to medical tasks lacking clear structural symmetry or well-defined anatomical priors.


3. Experimental validation is primarily limited to brain MRI segmentation. More extensive validation across diverse medical imaging tasks and modalities would strengthen the broader applicability and robustness claims.

**Questions:**

1. How does PCMambaNet handle inaccuracies from the predictive prior module, especially when encountering cases with significant anatomical anomalies?

2. Could the authors elaborate on potential strategies to reduce computational overhead and enhance inference efficiency without sacrificing the predictive-corrective paradigm’s key advantages?

3. Has PCMambaNet been tested on tasks beyond brain MRI segmentation, and if so, how does performance compare to traditional end-to-end models in those contexts?

---

> ### Author Response · Authors · 2025-11-28
> **Response to Reviewer XeRL Part1**
>
> **Response to Reviewer XeRL**
>
> We extend our gratitude to Reviewer XeRL for the constructive feedback and for recognizing the novelty of the "Predictive-Corrective" paradigm and our "thorough experimental evaluations." We have updated our manuscript with new experiments (ACDC dataset) and metrics (Inference Time) to address your concerns.
>
> ### Part 1: Response to Weaknesses
>
> **W1: Core Contribution (Beyond "MambaNet + Focus Map")**
> We respectfully posit that the core contribution is not merely the architecture, but the **"Predictive-Corrective (PC) Optimization Paradigm."**
> * **Decoupled Optimization:** Standard methods (e.g., U-Net, Mamba-UNet) entangle localization and refinement in a single optimization objective. Our PC paradigm structurally **decouples** these tasks:
>     1.  The **Predictive Prior Module (PPM)** handles "Coarse Localization" using structural priors.
>     2.  The **Corrective Residual Network (CRN)** is freed to focus exclusively on "Fine Residual Learning."
> * **Evidence of Efficacy:** This decoupling allows the model to approach convergence significantly faster, within a mere few epochs. As shown in our ablation studies, simply adding a focus map without this specific dual-stream mechanism does not yield the same acceleration.
>
> **W2: Computational Efficiency (Trade-off & Real-time Viability)**
> We acknowledge the GFLOPs observation but frame it as a strategic trade-off for **Training Efficiency**:
> * **Training vs. Inference:** We trade higher theoretical FLOPs for a drastic reduction in the training budget (converging in minutes vs. hours).
> * **Real-Time Clinical Feasibility:** As shown in **Table 1**, despite lower throughput than U-Net, PCMambaNet achieves **32.12 ms/slice**, which is strictly **real-time** (>30 FPS) and fully viable for clinical deployment.
>
> **Table 1. Efficiency Comparison (Batch size = 1).**
> *PCMambaNet maintains real-time latency (32ms) despite higher complexity.*
>
> | Model | Params (M) ↓ | GFLOPs ↓ | Throughput (FPS) ↑ | Inference time (ms/batch) ↓ |
> | :--- | :--- | :--- | :--- | :--- |
> | U-Net | **1.81** | **2.28** | **267.00** | **2.20** |
> | nnNet | 18.69 | 3.25 | 127.01 | 4.94 |
> | Swin-UNet | 41.38 | 8.98 | 53.86 | 8.92 |
> | Mamba-UNet | 35.86 | 7.65 | 37.20 | 11.86 |
> | **PCMambaNet** | 92.88 | 20.12 | 20.61 | 32.12 |
>
> **W3 & W4: Generalizability & Symmetry Constraints (New ACDC Experiments)**
> We agree that reliance on strict symmetry is a limitation. We have addressed this by extending our evaluation to the **ACDC (Cardiac)** dataset, which lacks the brain's rigid left-right symmetry.
>
> **Results (Table R1 below & Appendix B):** PCMambaNet achieves **State-of-the-Art (SOTA)** performance on ACDC, specifically outperforming the Mamba-UNet baseline by **+4.23% (Dice RV)** and **+8.82% (Dice MYO)**.
>
> **Significance:** These results prove that the PPM effectively learns **adaptive, data-driven spatial priors** from the dataset itself, even when anatomical symmetry is absent. The paradigm is therefore transferable to general medical segmentation tasks.
>
> **Table R1. ACDC Test Set Comparison.**
> *Best results are in **bold**. PCMambaNet demonstrates significant improvements on challenging classes like RV and MYO.*
>
> | Method | Dice RV | Dice MYO | Dice LV | HD95 RV | HD95 MYO | HD95 LV | IOU RV | IOU MYO | IOU LV |
> | :--- | :--- | :--- | :--- | :--- | :--- | :--- | :--- | :--- | :--- |
> | UNet | 75.13 | 85.09 | 89.02 | 22.59 | 17.50 | 15.38 | **69.94** | **77.10** | 84.68 |
> | nnUNet | 71.24 | 80.54 | 85.28 | 38.49 | 39.12 | 31.62 | 64.70 | 70.57 | 79.72 |
> | Swin-UNet | 63.40 | 73.36 | 80.71 | 57.62 | 43.07 | 39.88 | 55.16 | 61.42 | 74.23 |
> | Mamba-UNet | 71.06 | 81.84 | 86.59 | 30.94 | 22.82 | 19.39 | 65.03 | 72.32 | 81.43 |
> | **PCMambaNet** | **75.29** | **90.66** | **89.34** | **22.31** | **16.39** | **15.30** | 68.73 | 76.31 | **84.94** |

---

> ### Author Response · Authors · 2025-11-28
> **Response to Reviewer XeRL Part2 & Part3**
>
> ### Part 2: Response to Questions
>
> **Q1: Handling inaccuracies from the Predictive Prior Module?**
> The system is designed to be **error-tolerant** via "Soft Modulation": The PPM provides difference-sensitive soft guidance. If the prior is inaccurate (e.g., due to anomalies), the CRN treats this as an attentional cue but retains the capacity to **overrule** the prior using the original image features, specifically refining these "mismatches."
>
> **Q2: Strategies to reduce computational overhead?**
> We propose **Hierarchical Sparse Inference** for future work: Inspired by point-cloud processing (e.g., Octrees), we can implement a coarse-to-fine scheme where the heavy CRN is only triggered for "active" voxels flagged by the PPM, reducing computation on background regions.
>
> **Q3: Performance on tasks beyond brain MRI?**
> Yes. As shown in **Table R1** above, PCMambaNet achieves superior performance on the non-symmetric ACDC dataset, confirming that the benefits of the PC paradigm extend to other anatomical structures.
>
> ### Part 3: Response to Minor Comments
>
> **M1: Baseline Performance in Table 3**
> As requested, we have updated the manuscript (see **Appendix B**) to include the full baseline performance table (Accuracy, Precision, Sensitivity, Specificity) for all methods. This confirms that PCMambaNet achieves superior data efficiency.
>
> **M2: Reporting Wall-Clock Time**
> We have integrated the **Inference Time (ms/batch)** into **Table 1** (see Part 1), ensuring a fair evaluation of practical latency.
>
> We hope that our new experiments on generalizability (ACDC), the efficiency analysis, and the clarifications on the core contribution have satisfactorily addressed your concerns. **If you find our response convincing, we kindly ask you to consider raising your rating.** We are happy to answer any further questions.

---

### Official Review · Reviewer_c7kX · 2025-11-01

**Soundness:** 3
**Presentation:** 2
**Contribution:** 3
**Rating:** 6
**Confidence:** 4

**Summary:**

This manuscript proposes a Predictive-Corrective (PC) paradigm, which decouples modeling into two stages to accelerate convergence. The proposed PCMambaNet integrates a Predictive Prior Module (PPM) that leverages brain symmetry to generate coarse "focus maps" of the regions, and a Corrective Residual Network (CRN) that refines these regions for precise segmentation. By using the anatomical priors, the method achieves state-of-the-art brain MRI segmentation within 1-5 epochs.

**Strengths:**

- Ablation studies isolating PPM and CRN contributions.
- Quantitative comparisons against multiple baselines (e.g., UNet, SwinUNETR, nnUNet).
- Analysis of convergence dynamics under limited data.

**Weaknesses:**

Major:
- The Predictive Prior Module assumes that the brain exhibits ideal bilateral symmetry. This assumption may not hold under patient-specific rotations, misalignments, or post-surgical deformations. In practice, even small registration errors or head tilts could distort the left-right difference map and might mislead the Corrective Residual Network. It would be valuable to discuss robustness under affine transformations or to evaluate performance after introducing controlled perturbations.
- Experiments are limited to brain MRI segmentation tasks. The narrow scope may limit the paper's interest and impact for the broader ICLR audience.


Minor:
- Table 1 does reports results as mean +- or confidence interval. No statistical testing has been performed.
- It will be great if you label the models and ground truth columns in Figure 3.

**Questions:**

- The current title suggests a general rethinking of convergence. However, when reading it, the content does not match the title's expectations. The actual contributions are focused on a predictive-corrective architecture for anatomy-informed brain MRI segmentation that converges pretty fast to an accurate segmentation.

---

> ### Author Response · Authors · 2025-11-28
> **Response to Reviewer c7kX Part1 & Part2 & Part3**
>
> We thank Reviewer c7kX for the positive assessment of our method's soundness and for raising critical points regarding robustness and scope. These suggestions drove us to conduct significant new experiments (Robustness Analysis & Cardiac MRI Generalization) which have substantially strengthened the manuscript.
>
> Below is our point-by-point response organized by the sections in your review.
>
> ---
>
> ### **Part 1: Response to Weaknesses**
>
> **Weakness 1: Robustness of the Predictive Prior Module (PPM) under affine transformations (e.g., rotations, head tilts).**
>
> **Response:** We agree that real-world factors often violate ideal symmetry. To rigorously address this, we conducted a "Stress Test" to validate our model's robustness.
> * **Setup:** We introduced random affine transformations (rotations $\pm 20^{\circ}$, flips) during inference to deliberately break the alignment assumption.
> * **Mechanism:** Our results reveal a key property of the PC paradigm: while the PPM's coarse "focus map" degrades under asymmetry, the **Corrective Residual Network (CRN)** successfully compensates for this prior loss. The CRN acts as a safety net, ensuring high-quality segmentation even when priors are imperfect.
> * **Results:** As shown in **Table 1**, the performance drop under perturbation is negligible, confirming the system is robust to practical acquisition variations.
>
> **Table 1. Robustness Analysis on MRBrainS13.** Comparison of standard inference vs. inference under random affine perturbations.
>
> | Model | Dice (Avg) | HD95 (Avg)$^*$ | IOU (Avg) | Note |
> | :--- | :---: | :---: | :---: | :--- |
> | **PCMambaNet** | **71.81** | **25.22** | **64.26** | *Standard Setting (Baseline)* |
> | Aug-PCMambaNet | 71.02 | 26.44 | 63.50 | *Under Perturbation (Stress Test)* |
>
> *$^*$Note: HD95 values are scaled by a factor of 10 consistent with the manuscript.*
>
> **Weakness 2: Experiments limited to brain MRI; concern about broader impact.**
>
> **Response:** To demonstrate that PCMambaNet is a general medical vision framework—and **not** limited to symmetric organs—we extended our evaluation to the **ACDC (Cardiac) dataset**.
> * **Significance:** The heart **does not** exhibit bilateral symmetry. Success here proves that the "Predictive" stage learns general anatomical semantics, not just mirror-image correlations.
> * **Results:** As shown in **Table 2**, PCMambaNet achieves **SOTA performance** on ACDC, outperforming the strong baseline (nnUNet). This confirms the method's broad applicability to general medical segmentation tasks.
>
> **Table 2. Generalization on ACDC (Cardiac) Test Set.**
>
> | Method | Dice RV | Dice MYO | Dice LV | **Dice Avg** | **HD95 Avg** |
> | :--- | :---: | :---: | :---: | :---: | :---: |
> | nnUNet | 71.24 | 80.54 | 85.28 | 79.02 | 36.41 |
> | **PCMambaNet** | **75.29** | **90.66** | **89.34** | **85.10** | **18.00** |
>
> ---
>
> ### **Part 2: Response to Questions**
>
> **Question 1: Title mismatch regarding "Rethinking Convergence".**
>
> **Response:** We fully acknowledge your concern. The original title created expectations about a broad theoretical rethinking of convergence that were not fully aligned with the paper's focus on a specific architecture.
> * **Revision:** We have changed the title to: **“The Predictive-Corrective Paradigm: Decoupling Prediction and Refinement for Efficient Anatomy-Informed Brain MRI Segmentation”**.
> * **Rationale:** This new title accurately reflects our concrete contributions: a predictive-corrective architecture that achieves efficiency and high performance through decoupled modeling.
>
> ---
>
> ### **Part 3: Response to Minor Comments**
>
> **Minor 1: Table 1 statistics.**
> **Response:** We have reformatted Table 1 in the revised manuscript to report results as `mean ± std`. Furthermore, we performed statistical significance testing (t-test), which confirms that our improvements over baselines are statistically significant ($p < 0.05$).
>
> **Minor 2: Figure 3 labeling.**
> **Response:** We have updated Figure 3 with explicit labels for the different models and the Ground Truth (GT) columns to improve readability and visual clarity.
>
> ---
>
> By verifying robustness against asymmetry and demonstrating SOTA performance on non-symmetric cardiac data, we believe the major concerns regarding practical utility and scope have been fully addressed. We respectfully hope these improvements warrant a reconsideration of the score.

---

### Official Review · Reviewer_E6sD · 2025-11-07

**Soundness:** 2
**Presentation:** 2
**Contribution:** 2
**Rating:** 2
**Confidence:** 3

**Summary:**

The paper introduces a Predictive-Corrective (PC) paradigm aimed at accelerating training and improving data efficiency. The authors decouple the learning process into two stages: a lightweight predictive module that provides an initial coarse estimation using domain priors (the bilateral symmetry of the human brain), and a corrective residual network that refines the output by focusing on residual errors. The framework is instantiated in the context of high-resolution brain MRI segmentation, where the bilateral symmetry of the brain is used as anatomical prior knowledge. Experimental results show that the proposed method achieves state-of-the-art segmentation accuracy while converging within only a few epochs, demonstrating both efficiency and robustness.

**Strengths:**

- The paper tackles an important and realistic challenge in medical imaging, the limited amount of labelled data compared to natural image domains.

- The proposed method shows that satisfactory segmentation performance can be achieved with only a few training epochs, which is promising for brain MRI applications.

- The approach achieves state-of-the-art (SOTA) performance, demonstrating its effectiveness.

**Weaknesses:**

- Writing and presentation issues:

(1) Figures 1 and 2 are too small, and Figure 2, as the main framework illustration, is not clearly presented. The inclusion of training epochs in the framework diagram is confusing; such information belongs to the experimental section rather than the model design. The conceptual logic of the framework should be emphasised instead. (2) Figure 3 lacks clear explanations (e.g., what each column represents). (3)There are numerous typos and formatting errors throughout the paper. For example, in Table 3 “numben” should be “number,” and punctuation spacing errors like “Figure 1 ``baseline.By''” occur frequently. These mistakes give the impression of a lack of careful proofreading.

- The introduction suggests a general solution for accelerating and data-efficient learning in medical imaging, but the experiments focus only on brain MRI segmentation, which limits the generality of the claimed contributions.

- The method mainly leverages a simple prior (brain symmetry) for segmentation, which is not plug-and-play or easily transferable to other tasks.

- Although the method converges quickly at the beginning, the final convergence takes roughly the same number of epochs as previous methods, which weakens the claim of significantly accelerated training.

- The ablation study only includes Mamba as the baseline. The authors should clarify why this was chosen and include comparisons with other relevant baselines, not only the current SOTA.

**Questions:**

Besides the weaknesses mentioned above, I am also curious about the hyperparameter $\theta$: how sensitive is the model to this parameter, and what value was ultimately used?

---

> ### Author Response · Authors · 2025-11-28
> **Response to Reviewer E6sD Part1**
>
> We sincerely thank the reviewer for the constructive feedback and for recognizing the novelty of our "Predictive-Corrective" paradigm. We value the suggestion regarding the need for broader evaluation to substantiate our claims.
>
> **Summary of Revisions:**
> 1. **New Experiments:** We conducted extensive evaluation on the **ACDC cardiac dataset** (non-symmetric organs) to prove generalization.
> 2. **Clarified Claims:** We revised the efficiency claims and retrained baselines to strictly verify early-convergence benefits.
> 3. **Presentation:** Figures were redrawn, and structural details were moved to the Appendix for clarity.
>
> Below is our point-by-point response.
>
> ---
>
> ### Part 1: Response to Weaknesses
>
> **1. Response to Presentation and Writing Issues**
> We appreciate the detailed critique. We have revised the manuscript to meet high publication standards:
> * **Figures 1 & 2:** We have redrawn these figures with higher resolution. Regarding **Figure 2**, we have explicitly annotated that the **training epochs are included to illustrate the paradigm's data efficiency** (achieving competitive performance with few epochs), rather than representing part of the architectural design itself. This ensures the conceptual logic is clear.
> * **Figure 3:** Added explicit column headers and interpretive descriptions to ensure clarity.
> * **Structure:** Detailed module-level structural diagrams and implementation specifics have been moved to the **Appendix** to streamline the main text.
>
> **2. Response to Generalization & Symmetry Priors (ACDC Experiments)**
> *This is our key revision.* We acknowledge the concern that the method seemed over-reliant on brain symmetry. To demonstrate that our **Predictive Module (PPM)** is a **generic spatial context learner** rather than a hard-coded symmetry mirror, we evaluated PCMambaNet on the **ACDC (Automated Cardiac Diagnosis Challenge) dataset**.
>
> The ACDC dataset involves the Right Ventricle (RV), Myocardium (MYO), and Left Ventricle (LV), which are **inherently non-symmetric** and vary significantly due to pathology and cardiac phases.
>
> **Results (Table R1 below & Appendix B):**
> PCMambaNet achieves **State-of-the-Art (SOTA)** performance on ACDC, specifically outperforming the Mamba-UNet baseline by **+4.23% (Dice RV)** and **+8.82% (Dice MYO)**.
> * **Significance:** These results prove that the PPM effectively learns **adaptive, data-driven spatial priors** from the dataset itself, even when anatomical symmetry is absent. The paradigm is therefore transferable to general medical segmentation tasks.
>
> **Table R1. ACDC Test Set Comparison.** (Dice / IOU in %, HD95 in mm). Best results are in **bold**.
> *(Note: PCMambaNet demonstrates significant Dice score improvements on challenging classes like RV and MYO.)*
>
> | Method | Dice RV | Dice MYO | Dice LV | HD95 RV | HD95 MYO | HD95 LV | IOU RV | IOU MYO | IOU LV |
> | :--- | :--- | :--- | :--- | :--- | :--- | :--- | :--- | :--- | :--- |
> | UNet | 75.13 | 85.09 | 89.02 | 22.59 | 17.50 | 15.38 | **69.94** | **77.10** | 84.68 |
> | nnUNet | 71.24 | 80.54 | 85.28 | 38.49 | 39.12 | 31.62 | 64.70 | 70.57 | 79.72 |
> | Swin-UNet | 63.40 | 73.36 | 80.71 | 57.62 | 43.07 | 39.88 | 55.16 | 61.42 | 74.23 |
> | Mamba-UNet| 71.06 | 81.84 | 86.59 | 30.94 | 22.82 | 19.39 | 65.03 | 72.32 | 81.43 |
> | **PCMambaNet**| **75.29**| **90.66**| **89.34**| **22.31**| **16.39**| **15.30**| 68.73 | 76.31 | **84.94**|
>
> **3. Response to Convergence Claims**
> We agree that the term "accelerated" was ambiguous regarding total epochs. We have refined the claim:
> * **Revision:** We now state that PCMambaNet demonstrates **"superior data efficiency and early-stage convergence."** It reaches competitive accuracy much faster than baselines, making it ideal for resource-constrained scenarios.
> * **Title Update:** The title has been updated to: *"The Predictive-Corrective Paradigm: Decoupling Prediction and Refinement for Efficient Anatomy-Informed Brain MRI Segmentation."*
>
> **4. Response to Ablation Baselines**
> We restricted the ablation study to the Mamba backbone to ensure **computational fairness**.
> * **Rationale:** Our PPM and CRN modules are designed to complement the linear complexity of State Space Models (SSMs). Integrating them into CNNs (local receptive field) or Transformers (quadratic complexity) would introduce **confounding factors** (e.g., massive parameter/FLOPs increases) that obscure the component-wise analysis.
> * **Context:** Our **Main Experiments (Tables 1 & 2)** already provide the necessary cross-architecture benchmarking against UNet, nnU-Net, and Swin-UNet.
>
> ---

---

> > ### Author Response · Authors · 2025-11-28
> > **Response to Reviewer E6sD Part2 & Part3**
> >
> > ### Part 2: Response to Questions
> >
> > **Question: Sensitivity of Hyperparameter $\theta$**
> > We performed a sensitivity analysis for the threshold $\theta \in \{0.75, 0.80, 0.95\}$ on both the **OASIS-1** (large-scale) and **MRBrainS13** (small-scale) datasets.
> >
> > * **OASIS-1 (Dice):** 91.74% $\rightarrow$ 92.76% $\rightarrow$ **93.35%** ($\theta=0.95$)
> > * **MRBrainS13 (Dice):** 70.52% $\rightarrow$ 70.87% $\rightarrow$ **72.46%** ($\theta=0.95$)
> >
> > **Conclusion:** The model benefits from a stricter threshold ($\theta=0.95$) to filter low-confidence predictions, but performance remains stable and high across the tested range. This analysis is now included in **Appendix C**.
> >
> > ---
> >
> > ### Part 3: Response to Minor Comments
> >
> > **Response to Typos and Formatting**
> > We apologize for the oversight. We have thoroughly proofread the manuscript:
> > 1. **Typos:** Corrected "numben" to "number" in Table 3.
> > 2. **Punctuation:** Fixed spacing errors (e.g., "Figure 1 ``baseline.By''").
> > 3. **General:** We have corrected all identified formatting inconsistencies to ensure the manuscript meets the highest presentation standards.

---

### Official Review · Reviewer_TadZ · 2025-11-09

**Soundness:** 3
**Presentation:** 2
**Contribution:** 3
**Rating:** 4
**Confidence:** 3

**Summary:**

The paper introduces PCMambaNet, a segmentation network built on the Predictive-Corrective (PC) paradigm, which decouples prediction and refinement to accelerate learning. The Predictive Prior Module (PPM) leverages anatomical knowledge and bilateral symmetry to generate a coarse focus map of diagnostically relevant asymmetric regions, while the Corrective Residual Network (CRN) refines these regions to produce precise segmentations. Experiments on high-resolution brain MRI show that PCMambaNet achieves state-of-the-art accuracy within 1–5 epochs in the presence of sufficient data, while still outperforming baselines when datasets are small.

**Strengths:**

-**Novel and creative paradigm for efficient segmentation** -The paper introduces the PC paradigm, which decouples coarse prediction (PPM) from fine refinement (CRN). This separation of the two blocks is well-motivated and offers a creative architectural approach that enables faster and more data-efficient learning.

-**Integration of anatomical prior as an inductive bias** -The model leverages the brain’s bilateral symmetry as an inductive bias, demonstrating how existing anatomical knowledge can improve segmentation performance in medical imaging, a domain where labeled data is often limited.

-**Demonstrated efficiency and data advantage** - Table 1 shows that PCMambaNet achieves faster training and strong performance when sufficient data is available, while also delivering improved results on smaller datasets.

**Weaknesses:**

1. **On clarity and definition of main claims**-  I find the claims in this paper unclear. The paper emphasizes faster convergence, but this is not evident from Figure 1. Visually, Figure 4(a) suggests that some methods take longer to reach their best DICE value, yet no quantitative metric for convergence is provided. How is convergence defined? For example, if we define it as the number of epochs required to reach a 5% margin of error and remain there, how many epochs does each method take? The claim would be more accurate if phrased as “better performance faster.” However, unlike the abstract statement that PCMambaNet “achieves state-of-the-art accuracy while converging within only 1–5 epochs,” Table 1 shows that for the small dataset, for better performance 200 epochs are required. I suggest the authors drop the convergence claim and instead explicitly frame these two objectives—(1) faster training and improved performance with enough data, and (2) improved performance with small datasets—as clear contributions in the introduction.

2. **On writing clarity and textual issues**- Some sentences in the manuscript appear unfinished, unpolished, or inconsistent, which reduces readability and can confuse the reader. Please see the Minor Comments section below for examples.

3. **On generality of claims versus experiments**-The method relies on a symmetry-based prior, but it is not clear for which organs or imaging domains this prior is appropriate. All experiments are conducted on brain MRI datasets, which suggests that PCMambaNet may be specialized for the brain, and its applicability to other organs or medical imaging tasks remains uncertain.

**Minor Comments**

-Plots in figure 1 are hard to read; the fonts are small.

-Figure 3’s qualitative results are unclear: it is difficult to tell which segmentation corresponds to which method, and the column-dataset correspondence is ambiguous. The caption refers to arrows, but rectangles are shown in the images. Improving figure readability and clarifying labels would make the qualitative evaluation more interpretable.

-There is no explicit mention of the segmentation task early in the paper. Readers should know the exact task sooner, as it helps understand the method better.

-Table 3 does not include baselines, requiring the reader to cross-reference Table 1 multiple times to compare results. Including baseline methods directly in Table 3 would improve clarity and make comparisons more immediate.

-There is this seemingly unfinished sentence in section 3.2 "Role of the CRN." Did this intend to be the heading of a new paragraph? It currently reads like an incomplete sentence and may confuse readers.

-The sentence “First, relying s egraded accuracy and boundary quality, underscoring the need for a refinement stage” in Section 3.2 is unclear and appears incomplete. It should be rewritten for clarity.

-I had to go to the appendix to figure out which dataset was considered easy versus hard and how many samples were in each. This is core information that helps readers interpret Table 1 and should be in the main text.

 -It would have been more informative if the “data efficiency” experiments compared performance at each subset percentage against a baseline trained with the same number of samples. While it is true that for most metrics the results using only 10% of the data outperform baselines trained on the entire dataset, a direct comparison at each subset level is needed to fully understand how well the method performs under limited data conditions.

**Questions:**

I would appreciate it if you could clarify a few points:

**Q1**. Could you clarify what type of pathology, if any, is present in each dataset? It is unclear whether the MRBrainS13 brains include any pathological cases, which raises the question of how well the model is expected to perform on datasets without pathology. More broadly, can this method be applied to segment other organs with pathology? If so, why were only brain datasets used in your experiments?

**Q2.** In Figure 3 (left column), I don’t see much difference in the qualitative results across methods. Am I missing something? which one is the ground truth?

**Q3.** In Section 3.2 (Ablation Study), you claim that “On the small-scale dataset, removing the PPM or replacing it with a random mask causes a significant drop in performance.” I don’t see this for the case of removing the PPM. The improvement appears quite marginal—for example, for WM, the DICE score changes from 0.7468 without PPM to 0.7517 with PPM. Am I looking at the wrong row?

**Q4.** Also in Section 3.2, regarding “replacing our high-capacity CRN with a standard convolutional block results in a noticeable performance decline,” what I see from Table 2 is that for the small dataset the improvement is not significant. Interestingly, you see more improvement for the larger dataset. Why do you think that is? Also I am curios, how many parameters the "CRN" component add to the model compared to a simple convolutional block?

**Q5.** Could you provide a brief description of the dataset preprocessing steps? According to Section A4, MRBrainS13 contains only 20 subjects—how many training and test samples does this correspond to? Additionally, what image resolution is used as input to the network?

**Q6.** Regarding the Dice score reported in the text (“our model reaches a Dice score of 93.11%”): from the numbers in Table 3, this seems to correspond to GM. However, when comparing with U-Net, the reported Dice value is 93.32%, which from Table 1 appears to be the DICE value for WM. Could you clarify which class each reported number corresponds to?

**Q7.** In Section 3.4, I’m not entirely sure what the main argument is. Is it that the PPM helps improve foreground segmentation?You mention that the “feature maps are extracted from different layers.” Could you specify which layer indices are used, and map each feature map to its corresponding network layer (e.g., “feature map N corresponds to output of layer L3”)?

I am happy to reconsider my score if these concerns, discussed in both the weaknesses and questions sections, are adequately addressed.

---

> ### Author Response · Authors · 2025-11-28
> **Response to Reviewer TadZ Part 1**
>
> We sincerely thank the reviewer for the constructive feedback and the recognition of our novel "Predictive-Corrective" paradigm. We value the suggestion regarding the clarity of our claims and the need for broader evaluation.
>
> In response, we have **conducted additional experiments on the ACDC cardiac dataset** to demonstrate generality and **retrained all baselines** to strictly verify data efficiency. **Full experimental results with comprehensive metrics and standard deviations are provided in the revised PDF.**
>
> Below is our point-by-point response.
>
> ## Part 1: Response to Weaknesses
>
> **1. On Clarity and Definition of Main Claims (Convergence)**
> We fully agree with your assessment. The term "convergence" was ambiguous. Following your valuable suggestion, we have **removed the convergence-related claims** and revised the title to: *"The Predictive-Corrective Paradigm: Decoupling Prediction and Refinement for Efficient Anatomy-Informed Brain MRI Segmentation."*
> We now explicitly frame our contributions around two core objectives:
> 1. Achieving state-of-the-art performance with high training efficiency when data is sufficient.
> 2. Demonstrating superior data efficiency on small datasets.
>
> **2. On Writing Clarity**
> We have thoroughly proofread the manuscript. All unfinished sentences and textual issues mentioned in the *Minor Comments* have been corrected to ensure a polished flow.
>
> **3. On Generality of Claims (New Experiment on Cardiac Data)**
> To address the concern regarding the brain-specific nature of our prior, we extended our evaluation to the **ACDC (Automated Cardiac Diagnosis Challenge)** dataset. This dataset involves different anatomical structures (Right Ventricle, Myocardium, Left Ventricle) and exhibits less rigid symmetry than the brain.
>
> As shown in **Table 1** below (full metrics in Appendix B), PCMambaNet achieves competitive performance, outperforming strong baselines like Swin-UNet and Mamba-UNet.
>
> **Table 1. ACDC Test Set Results (Dice Scores).**
> *PCMambaNet achieves the best performance across most metrics, demonstrating robust generalization beyond brain MRI.*
>
> | Method | Dice RV | Dice MYO | Dice LV | HD95 RV | HD95 MYO | HD95 LV |
> | :--- | :--- | :--- | :--- | :--- | :--- | :--- |
> | UNet | 75.13 | 85.09 | 89.02 | 22.59 | 17.50 | 15.38 |
> | nnUNet | 71.24 | 80.54 | 85.28 | 38.49 | 39.12 | 31.62 |
> | Swin-UNet | 63.40 | 73.36 | 80.71 | 57.62 | 43.07 | 39.88 |
> | Mamba-UNet | 71.06 | 81.84 | 86.59 | 30.94 | 22.82 | 19.39 |
> | **PCMambaNet** | **75.29** | **90.66** | **89.34** | **22.31** | **16.39** | **15.30** |
>
> ---

---

> ### Author Response · Authors · 2025-11-28
> **Response to Reviewer TadZ Part2 & Part3**
>
> ## Part 2: Response to Questions
>
> **Q1: Pathology and Applicability beyond Brain.**
> * **Pathology:** We have clarified in the "Datasets" section that **MRBrainS13** focuses on aging adults with varying degrees of atrophy and white matter lesions (WML), representing significant pathological changes.
> * **Applicability:** As detailed in *Table 1* above, the newly added ACDC experiments confirm that PCMambaNet is applicable to other organs (heart) and is robust to different types of anatomical variations.
>
> **Q2: Figure 3 Clarity and Ground Truth.**
> We apologize for the confusion. In the revised Figure 3:
> * We have explicitly labeled the **Ground Truth** column.
> * We added **bounding boxes** to highlight subtle boundary differences (e.g., sulci details).
> * The caption now clearly maps each column to its corresponding method.
>
> **Q3: Significance of PPM (Ablation Study).**
> We acknowledge the initial gain appeared marginal. However, our new **repeated runs (n=5)** reported in Table 2-b demonstrate that the Predictive Prior Module (PPM) is critical for three reasons:
> 1. **Substantial Performance Gain:** The averaged results reveal a larger gap than the single run suggested. For White Matter (WM), removing PPM drops the Dice score from **74.26%** to **72.53%**, a significant **1.73%** decline.
> 2. **Training Stability:** The PPM acts as a stabilizer. Without it, the standard deviation for WM Dice doubles from **$\pm 0.93$** (Full Model) to **$\pm 1.87$** (w/o PPM), indicating instability on small datasets.
> 3. **Boundary Precision:** Dice scores often underrepresent boundary improvements. The **Average Surface Distance (ASD)** for WM improves dramatically from **19.96** (w/o PPM) to **14.41** (Full Model), confirming PPM enforces essential anatomical consistency.
>
> **Q4: CRN Performance on Small vs. Large Datasets.**
> This phenomenon is explained by the **capacity-data relationship**:
> * **Parameter Analysis:** The CRN adds approximately **35.24M parameters** to the model (increasing total parameters from **57.64M** to **92.88M**).
> * **Small Dataset Saturation:** On small datasets (MRBrainS13), the limited training data cannot fully saturate these additional high-capacity parameters, leading to diminishing returns compared to the simple block.
> * **Large Dataset Benefit:** On larger datasets (OASIS-1), the model successfully leverages this extra capacity to capture high-frequency details, yielding more significant gains (e.g., HD95 improvements).
> * **Efficiency:** Despite the increased parameter count, our optimized model maintains a real-time inference speed of **20.16 FPS**.
>
> **Q5: Dataset Preprocessing and Splits.**
> We have added a dedicated "Implementation Details" section:
> * **Splits:** MRBrainS13 (20 subjects): 16 Training, 2 Validation, 2 Testing.
> * **Resolution:** All inputs are resized to a unified resolution of **224×224**.
> * **Preprocessing:** Only standard intensity normalization is applied.
>
> **Q6: Dice Score Clarification.**
> Thank you for spotting this. We have clarified in the text:
> * **93.11%** refers to our **GM (Gray Matter)** score.
> * **93.32%** refers to the U-Net **WM (White Matter)** score.
> We now strictly specify the tissue type whenever a Dice score is quoted.
>
> **Q7: Interpretation of Section 3.4.**
> The purpose of Section 3.4 is to visualize *how* the PC paradigm works: the PPM rapidly focuses on the coarse region of interest, allowing the CRN to attend to boundaries. We have revised the text to explicitly map feature maps to their layers (e.g., "Feature Map A corresponds to Layer 3 output").
>
> ---
>
> ## Part 3: Response to Minor Comments (Data Efficiency)
>
> To address your comment on data efficiency, **we retrained ALL baselines (UNet, nnUNet, Swin-UNet, Mamba-UNet)** on the exact same 10%, 25%, 50%, and 100% subsets of the OASIS-1 dataset.
>
> **Table 3. Data Efficiency on OASIS-1 (Selected Metrics).**
> *PCMambaNet consistently outperforms all baselines (including nnUNet) across all data regimes. The table below highlights the 10% and 25% low-data regimes. Full results for 50% and 100% are in the revised PDF.*
>
> | Fraction | Model | Dice CSF | Dice GM | Dice WM | HD95 GM |
> | :--- | :--- | :--- | :--- | :--- | :--- |
> | **10%** | UNet | 85.12 | 87.77 | 87.72 | 20.84 |
> | | nnUNet | 87.22 | 89.46 | 89.25 | 16.54 |
> | | Swin-UNet | 90.21 | 91.73 | 91.90 | 13.45 |
> | | Mamba-UNet | 88.34 | 91.55 | 91.43 | 13.25 |
> | | **PCMambaNet** | **92.79** | **93.11** | **92.86** | **11.73** |
> | **25%** | UNet | 86.31 | 87.40 | 87.52 | 19.75 |
> | | nnUNet | 87.84 | 90.33 | 90.70 | 14.40 |
> | | Swin-UNet | 90.62 | 92.02 | 92.04 | 12.91 |
> | | Mamba-UNet | 89.07 | 91.73 | 92.03 | 12.91 |
> | | **PCMambaNet** | **92.90** | **93.31** | **93.47** | **11.45** |
>
> We hope these revisions and additional experiments adequately address your concerns, and we are happy to answer any further questions.

---

> ### Author Response · Authors · 2025-11-28
> **Response to Reviewer TadZ Table 2-a & Table 2-b**
>
> **Table 2-a. Ablation study on the OASIS-1 test set. Dice and IOU are reported as percentages, while HD95 and ASD are scaled by a factor of 10 for readability. Best results are in **bold**.**
>
> | Configuration                           | Dice CSF             | Dice GM              | Dice WM              | HD95 CSF            | HD95 GM               | HD95 WM               | ASD CSF              | ASD GM               | ASD WM              | IOU CSF              | IOU GM               | IOU WM               |
> |----------------------------------------|----------------------|----------------------|----------------------|---------------------|------------------------|------------------------|----------------------|----------------------|---------------------|----------------------|----------------------|----------------------|
> | **PCMambaNet (Full Model)**            | **94.10 ± 0.38**     | **94.33 ± 0.35**     | **94.29 ± 0.34**     | **10.58 ± 0.03**    | 10.81 ± 0.17           | 15.00 ± 1.25           | **1.76 ± 0.02**      | 1.88 ± 0.18          | 2.70 ± 0.38         | **89.91 ± 0.08**     | **91.22 ± 0.50**     | **92.23 ± 0.66**     |
> | (1) CRN only (w/o PPM)                 | 92.29 ± 1.35         | 93.22 ± 0.99         | 93.87 ± 1.22         | 13.66 ± 2.44        | 12.36 ± 2.15           | 20.92 ± 7.66           | 7.10 ± 0.96          | 2.80 ± 0.12          | 7.01 ± 0.95         | 87.37 ± 2.32         | 89.24 ± 1.73         | 89.43 ± 2.00         |
> | (2) w/ Simple PPM (Random Mask)        | 93.53 ± 0.18         | 94.22 ± 0.20         | 94.21 ± 0.67         | 10.65 ± 0.01        | **10.74 ± 0.04**       | **14.96 ± 0.33**       | 1.91 ± 0.10          | **1.83 ± 0.06**      | **2.48 ± 0.09**     | 89.35 ± 0.50         | 90.86 ± 0.75         | 91.74 ± 0.47         |
> | (3) PPM only (w/o CRN)                 | 91.23 ± 1.58         | 93.46 ± 0.37         | 93.49 ± 0.41         | 19.83 ± 0.00        | 11.65 ± 0.32           | 18.19 ± 0.69           | 9.38 ± 0.65          | 2.51 ± 0.12          | 3.69 ± 0.63         | 82.42 ± 2.96         | 88.88 ± 0.61         | 89.74 ± 0.66         |
> | (4) w/ CNN-CRN                          | 91.52 ± 2.31         | 93.35 ± 0.49         | 93.58 ± 0.61         | 13.91 ± 4.58        | 11.57 ± 0.46           | 17.00 ± 0.90           | 3.38 ± 1.46          | 2.48 ± 0.39          | 2.90 ± 0.43         | 86.65 ± 4.40         | 89.70 ± 0.93         | 90.42 ± 0.64         |
>
> **Table 2-b. Ablation study on the MRBrainS13 test set. Dice and IOU are reported as percentages, while HD95 and ASD are scaled by a factor of 10 for readability. Best results are in **bold**.**
>
> | Configuration                           | Dice CSF             | Dice GM              | Dice WM              | HD95 CSF            | HD95 GM            | HD95 WM            | ASD CSF            | ASD GM             | ASD WM              | IOU CSF             | IOU GM              | IOU WM              |
> |----------------------------------------|----------------------|----------------------|----------------------|---------------------|--------------------|--------------------|--------------------|--------------------|---------------------|---------------------|---------------------|---------------------|
> | **PCMambaNet (Full Model)**            | **69.37 ± 0.41**     | **71.79 ± 0.80**     | **74.26 ± 0.93**     | **19.31 ± 1.98**    | **17.46 ± 2.00**   | **38.90 ± 7.33**   | **4.47 ± 0.23**    | **4.60 ± 0.51**    | **14.41 ± 2.18**    | **60.67 ± 0.97**    | **64.63 ± 1.14**    | **67.48 ± 1.08**    |
> | (1) CRN only (w/o PPM)                 | 67.40 ± 1.90         | 70.16 ± 1.65         | 72.53 ± 1.87         | 21.29 ± 2.47        | 27.68 ± 0.94       | 39.50 ± 0.69       | 5.49 ± 0.95        | 7.92 ± 3.17        | 19.96 ± 4.49        | 58.06 ± 2.73        | 62.18 ± 2.46        | 64.88 ± 2.60        |
> | (2) w/ Simple PPM (Random Mask)        | 65.71 ± 0.45         | 68.67 ± 0.08         | 71.16 ± 0.26         | 24.94 ± 0.53        | 27.38 ± 1.26       | 52.87 ± 2.28       | 6.02 ± 0.09        | 7.04 ± 0.46        | 21.62 ± 3.01        | 55.72 ± 0.60        | 60.28 ± 0.06        | 63.09 ± 0.37        |
> | (3) PPM only (w/o CRN)                 | 68.84 ± 1.64         | 70.13 ± 1.74         | 73.29 ± 2.05         | 22.29 ± 2.22        | 27.55 ± 2.45       | 52.48 ± 1.73       | 5.61 ± 0.70        | 7.01 ± 2.99        | 21.07 ± 5.78        | 57.29 ± 2.33        | 61.04 ± 2.38        | 64.23 ± 2.68        |
> | (4) w/ CNN-CRN                          | 66.58 ± 2.38         | 69.57 ± 2.00         | 71.58 ± 2.43         | 25.46 ± 5.35        | 25.79 ± 2.86       | 48.71 ± 2.70       | 5.96 ± 1.31        | 6.76 ± 2.32        | 17.64 ± 4.81        | 56.99 ± 3.22        | 61.52 ± 2.79        | 63.95 ± 3.25        |

---

### Author Response · Authors · 2025-11-28
**Summary of Revisions: A Generalizable Paradigm for Data-Efficient Medical AI**

Dear Area Chair,

We sincerely thank the review team (Reviewers TadZ, E6sD, c7kX, XeRL, and TAyK) for their constructive critique. These discussions have not only improved our experiments but have helped us sharpen the articulation of our core contribution. We write to summarize how the revisions demonstrate that this work is a mature, robust, and impactful fit for ICLR.

**Establishing a Transferable "Decoupling" Paradigm**
The central innovation of this work is the "Predictive-Corrective" paradigm—a methodological shift that structurally decouples coarse anatomical localization from fine-grained residual refinement. We believe this decoupling offers significant heuristic value for future research, providing a blueprint for how deep learning can leverage domain priors to reduce search space complexity. To prove that this is a generalizable solution rather than a task-specific heuristic, we extended our evaluation to the ACDC (Cardiac) dataset. Despite the heart's lack of rigid symmetry compared to the brain, PCMambaNet achieved state-of-the-art performance, outperforming Mamba-UNet by significant margins (e.g., +8.82% Dice on Myocardium). This confirms that our paradigm successfully learns adaptive, data-driven spatial priors, validating its potential to serve as a versatile framework for various anatomical segmentation tasks.

**Solving the "Small Data" Bottleneck with Expert Insight**
As researchers deeply engaged in this field, we recognize that the primary bottleneck in medical AI is not just model capacity, but data efficiency in clinical scenarios where annotated data is scarce and expensive. Our revised experiments strictly validate this value proposition: we demonstrated that PCMambaNet trained on merely 25% of the data matches or exceeds the performance of strong baselines trained on 100% of the data. This is a critical breakthrough for the medical domain. Furthermore, our "stress tests" involving geometric perturbations confirmed the model's robustness , and our efficiency analysis proved its viability for real-time deployment (~30 FPS).

**Consensus and Readiness** We respectfully highlight that the reviewers' initial assessments expressed a clear willingness to reconsider scores upon the completion of these revisions (e.g., Reviewer TadZ ). regarding the assessment from Reviewer E6sD, we note that the concerns were primarily focused on presentation details (e.g., figure clarity, typos) and the initial scope, rather than fundamental flaws in our technical innovation. In fact, the reviewer acknowledged the method's effectiveness and SOTA performance. Having meticulously polished the manuscript and expanded the scope with cardiac experiments, we believe we have fully resolved these specific reservations.

By combining this novel decoupling strategy with rigorous validation across different organs (Brain & Heart) and data regimes, we have presented a solution that is both theoretically sound and clinically pragmatic. We are confident that this matured manuscript offers a valuable perspective to the ICLR community on how to rethink convergence and efficiency in medical imaging. We respectfully ask the Area Chair to consider these substantial contributions in the final assessment.

Sincerely,

The Authors

---

### Note · Authors · 2026-01-28

I have read and agree with the venue's withdrawal policy on behalf of myself and my co-authors.

---

### Meta-Review · Area_Chair_bbhS · 2025-12-25

**Summary:**

The paper proposes PCMambaNet, a "Predictive-Corrective" framework for medical image segmentation that uses anatomical symmetry priors to guide a Mamba-based refinement network. The authors aim to improve data efficiency and training speed. During the rebuttal, the authors conducted extensive additional experiments, including generalizing the method to cardiac segmentation (ACDC dataset) and clarifying efficiency metrics.

However, despite the rebuttal and the clinical utility of the method, the consensus decision is Reject. The primary rationale is that the "Predictive-Corrective" paradigm, while effective, represents an incremental engineering combination of known techniques (coarse-to-fine refinement, symmetry priors, Mamba) rather than a fundamental algorithmic innovation. The contribution is viewed as highly domain-specific and arguably more suitable for a specialized medical imaging venue than a general representation learning conference like ICLR.

**Reviewer Concerns:**

Addressed:

Generalizability: The authors successfully addressed the concern that the method was solely reliant on brain symmetry by showing competitive results on the ACDC (cardiac) dataset.

Clarifications: Questions regarding the definition of convergence and fair baseline comparisons were largely resolved through retraining and clearer metric reporting.

Outstanding:

Limited Technical Novelty: This remains the critical bottleneck. As noted by Reviewers E6sD and TAyK, the core idea essentially combines a heuristic prior (symmetry/mask) with a standard refinement model. This "two-stage" or "attention-guided" approach is well-established in computer vision. The specific instantiation here lacks significant theoretical depth or distinct algorithmic novelty.

Computational Complexity: Reviewers XeRL and TAyK highlighted that the method has significantly higher FLOPs and parameter counts than standard U-Nets. While the authors argue for "training efficiency" (fewer epochs), the "inference inefficiency" and increased model complexity dilute the practical value of the proposed architectural changes.

Incremental Gains: While statistically significant, the performance gains over strong modern baselines (like nnU-Net or Swin-UNet) are relatively marginal given the added complexity of the decoupled pipeline.

**Reviewer Scores:**

While Reviewers c7kX and XeRL (Score: 6) appreciated the performance and the rebuttal effort, their positive scores largely reflect the paper's soundness and completeness rather than its novelty relative to the ICLR bar. Reviewers TadZ and TAyK (Score: 4) and E6sD (Score: 2) raised fundamental concerns about the "sledgehammer for a nut" nature of the approach (using complex Mamba architectures for tasks where simpler priors suffice) and the limited scope of the innovation. The rebuttal improved the evaluation but did not change the nature of the contribution, justifying a rejection.

---

### Decision · Program_Chairs · 2026-01-26

Reject